# Monoclonal humanized monovalent antibody blocking therapy for anti-NMDA receptor encephalitis

Atsuo Kanno[1,2,6], Takuya Kito [2,6], Masashi Maeda[1,2], Shanni Yamaki[1], Yasushi Amano [2], Takuya Shimomura [2], Margarita Anisimova[3], Naomi Kanazawa [4], Koichiro Suzuki[2], Amir Razai[1], Takuma Mihara[2], Kaori Kubo[2], Takeshi Shimada[2], Koji Nakamura[2], Naoko Nomura[2], Yuji Kondo[2], Akira Okimoto [2], Azusa Sugiyama[2], Deborah Park[3], Ivar Stein[3], Samuel Petshow[3], Valentin Vandendoren [5], Sanela Bilic[5], Roghiye Kazimi[1], Vallari Eastman[1], Scott J. Snipas[1], Mathew Mitchell [1], Mari Maurer[1], Marty Jefson[1], Jay Lichter[1], Daisuke Yamajuku[2], Hiroki Shirai [2], Megumi Adachi [2], Daniel J. Hoeppner [2], Satoshi Kubo[1,2], Karen Zito[3], Takahiro Iizuka [4], Peter Flynn[1] & Mitsuyuki Matsumoto [1,2] ✉

Anti-NMDA receptor (NMDAR) encephalitis is a devastating disease with severe psychiatric and neurological symptoms believed to be caused by pathogenic autoantibodies that bind to the N-terminal domain (NTD) of the NMDAR GluN1 subunit (GluN1-NTD) crosslinking adjacent NMDARs and driving their internalization. Here we describe ART5803, a humanized monovalent antibody, as a potential therapy for anti-NMDAR encephalitis. ART5803 binds with a high affinity ($K_D = 0.69$ nM) to GluN1-NTD without affecting NMDAR activity or inducing internalization. ART5803 blocks NMDAR internalization induced by patients' pathogenic autoantibodies, and restores NMDAR function. A marmoset animal model was developed using sustained intracerebroventricular (ICV) administration of a human pathogenic autoantibody to evoke behavioral and motor abnormalities. ART5803 ICV infusion or peripheral injections rapidly reversed these abnormalities. These data, together with the pharmacokinetic profile in cynomolgus monkeys, indicate a therapeutic potential for intravenous (IV)-administered ART5803 as a fast-acting and efficacious option for anti-NMDAR encephalitis.

Encephalitis is a serious condition associated with psychiatric, neurological, and movement disorder manifestations. There are many potential causes of encephalitis which can primarily be categorized as either infectious or autoimmune. Among encephalitis cases with autoimmune etiology, one of the most prevalent subtypes is anti-NMDA receptor (NMDAR) encephalitis[1], which has been increasingly reported since its first description in 2007 by Dalmau and colleagues[2,3]. The NMDAR is a glutamate receptor that forms a cation-selective channel found on excitatory synapses, playing a critical role in multiple neural functions including neuroplasticity, learning, memory, and motor function. Dysregulation of this pathway is associated with a number of psychiatric and neurological disorders

[1]Arialys Therapeutics, Inc., La Jolla, CA, USA. [2]Astellas Pharma Inc., Tsukuba, Ibaraki, Japan. [3]Center for Neuroscience, University of California, Davis, CA, USA. [4]Department of Neurology, Kitasato University School of Medicine, Sagamihara, Kanagawa, Japan. [5]Vanadro, LLC, Waukee, IA, USA. [6]These authors contributed equally: Atsuo Kanno, Takuya Kito. ✉e-mail: mmatsumoto@arialysrx.com

including dementia and psychosis related disorders such as schizophrenia[4,5].

Anti-NMDAR encephalitis is significantly more common in women than men, and a large proportion of patients are pediatric, with an median age of onset of 21 years[6,7]. Patients develop prominent psychiatric symptoms, altered consciousness, seizures, movement disorders including dyskinesia, and life-threatening dysautonomia including central hypoventilation[1]. While the cause for the immune system's generation of autoantibodies is not always known, 30-60% of adult female cases are associated with ovarian teratomas and there is also an association with herpes simplex encephalitis[7,8]. The autoantibodies can mediate disease progression[1,9,10] by targeting the GluN1 subunit of NMDAR, recognizing selective epitopes in its N-terminal domain (NTD). In anti-NMDAR encephalitis patients, pathogenic autoantibodies are exclusively of the bivalent IgG class, and their binding can induce cross-linking of adjacent NMDARs, leading to receptor internalization and an overall hypo-NMDAR status[11–13].

Up to 70% of patients with anti-NMDAR encephalitis are admitted to the intensive care unit (ICU) at some point during their disease progression[6], typically for airway protection, persistent dysautonomia, fluctuations in consciousness, or breathing dysfunction. Patients in the ICU receive prolonged courses of sedatives, antiepileptics, neuromuscular blockers, empirical antibiotics, and neuro- or psychoactive medications in an attempt to manage symptoms and prevent infections. Currently, there is no regulatory approved treatment for anti-NMDAR encephalitis. However, the consensus among treating physicians is that immunotherapy, and tumor removal in cases with ovarian teratomas, is the best course of action[6,7]. First-line standard-of-care includes steroids and intravenous immunoglobulins, with or without plasma exchange. Nonresponding patients may receive second-line therapies, typically rituximab or cyclophosphamide. For the remaining patients who are refractory to first- and second-line treatments, third-line options include bortezomib or tocilizumab.

While current patient outcomes are considered good with first- and second-line therapies, up to 10% of cases are fatal, and a significant percentage of patients (up to 28%) do not respond to therapies, and up to 12% of adults and 25% of children experience relapse[6,7]. The lack of a specifically approved therapy combined with significant fatality rate, morbidity, prolonged time to recovery, relapse rate, and long-term cognitive deficits, highlight the profound unmet medical need in this disease. Moreover, currently available therapies are broadly immunosuppressive and are associated with significant side effects, including infection risk. A more targeted approach may allow for a swifter and more effective recovery, with a more favorable safety profile in this population.

We have developed ART5803, a humanized monovalent IgG1 antibody which binds to NMDAR with a high affinity without affecting NMDAR activity or inducing internalization. We demonstrate that ART5803 prevents crosslinking driven internalization of NMDARs by pathogenic autoantibodies and enables the recovery of cell-surface NMDAR expression and functions in both cellular and neuronal models. Importantly, peripheral administration of ART5803 reverses behavioral abnormalities and restores reduction of NMDAR expression in a marmoset disease model, which demonstrates ART5803 as a potential therapy for anti-NMDAR encephalitis.

## Results
### Generation of ART5803, a humanized monovalent antibody that binds to the NTD of GluN1 subunit (GluN1-NTD) of NMDAR with high affinity

Previous publications describe the cloning of monoclonal antibodies from the CSF of anti-NMDAR encephalitis patients that are thought to be responsible for disease pathogenesis[13,14]. We therefore recombinantly synthesized representative pathogenic autoantibodies. Among them, autoantibody #003-102 (#003-102 Ab) showed the strongest

pathogenicity and was used as the primary patient-derived pathogenic autoantibody throughout the present study.

Previous reports suggest that a predominant mechanism driving anti-NMDAR encephalitis is the induction of NMDAR internalization triggered by NMDAR crosslinking by bivalent pathogenic anti-NMDAR autoantibodies[1,10,12]. These studies highlighted that reduced NMDAR signaling was due to receptor internalization rather than any direct antagonism by these pathogenic autoantibodies[11]. In addition, Fab fragments of the autoantibodies (monovalent fragments) do not induce receptor internalization or cause significant differences in NMDAR-mediated electrophysiology[11,12]. The antigen recognition sites of pathogenic autoantibodies have been previously reported to be restricted to the GluN1-NTD[15]. Therefore, we hypothesized that a monovalent antibody, that could bind to the GluN1-NTD and block the binding of bivalent pathogenic autoantibodies and crosslinking of NMDARs without impacting NMDAR function, could have therapeutic potential in the treatment of anti-NMDAR encephalitis (Fig. 1a).

On consideration of the molecular structure of a therapeutic, we decided to focus on a monovalent IgG. Although a Fab fragment could be a possible monovalent therapeutic form, the quick clearance of Fab fragments from systemic circulation relative to Fc-containing IgG structures has been well documented[16]. Further the relative bulkiness (100 KDa) of a monovalent IgG over a Fab was thought to result in greater potential steric hindrance to pathogenic autoantibodies.

To generate prototype antibodies binding to the GluN1-NTD with high affinities, BALB/cAJcl mice were immunized with human GluN1-NTD recombinant protein with adjuvants. Fluorescence-activated cell sorting (FACS) was used to isolate plasma/plasmablast cells specifically binding to human GluN1-NTD derived from immunized mice lymph nodes. Using a rapid antibody isolation system[17,18], which enables identification of more potent and unique antibodies than the conventional hybridoma methods, we selected 4 prototype antibodies with strong binding activities to the GluN1-NTD and strong inhibitory activities against pathogenic #003-102 Ab binding.

To create a monovalent antibody, we used knobs-into-hole technology[19]. "Hole" mutations (T366S / L368A / Y407V) were introduced in the CH3 domain of the parental antibody heavy chain. A "Knob" mutation (T366W) was introduced in the CH3 of the Fc protein and combined with the "Hole" introduced heavy chain. In order to avoid antibody-dependent cell-mediated cytotoxicity (ADCC), antibody-dependent cell-mediated phagocytosis (ADCP) or complement-dependent cytotoxicity (CDC), LALA mutations (L234A / L235A) were introduced in the CH2 domain[20] to reduce binding to the IgG Fc receptors FcγRI, FcγRII, and FcγRIII as well as to complement component C1q (Supplementary Figs. S1 and S2). These knobs-into-hole and LALA mutations have been successfully introduced to therapeutic monoclonal antibodies without increasing immunogenicity[21,22]. ART5803 was finally selected as a humanized IgG1 monovalent antibody having the highest affinity to the GluN1-NTD (Fig. 1b).

We performed surface plasmon resonance analysis to determine the affinity of ART5803 and the pathogenic #003-102 Ab against the human GluN1-NTD. ART5803 showed a stronger affinity ($K_D = 6.92 \times 10^{-10}$M) compared to #003-102 Ab ($K_D = 2.85 \times 10^{-7}$M) for GluN1-NTD (Supplementary Table S1). The ability of ART5803 to block the binding of #003-102 Ab was tested by competitive ELISA. ART5803 was able to compete with the binding of biotinylated #003-102 Ab (0.60 μg/mL) to human GluN1-NTD with an $EC_{50}$ of 0.30 μg/mL, which was much more potent than #003-102 Ab by itself (EC50 = 6.2 μg/mL) (Fig. 1c).

### ART5803 and a pathogenic autoantibody (#003-102 Ab) bind to shared epitopes on GluN1-NTD

We determined the crystal structure of the pathogenic #003-102 Ab Fab' in complex with the GluN1-NTD protein at 3.5Å resolution. The GluN1-NTD consists of two subdomains, R1 and R2[23]. As shown in Fig. 1d, e, #003-102 Ab bound to the conformational epitope on the

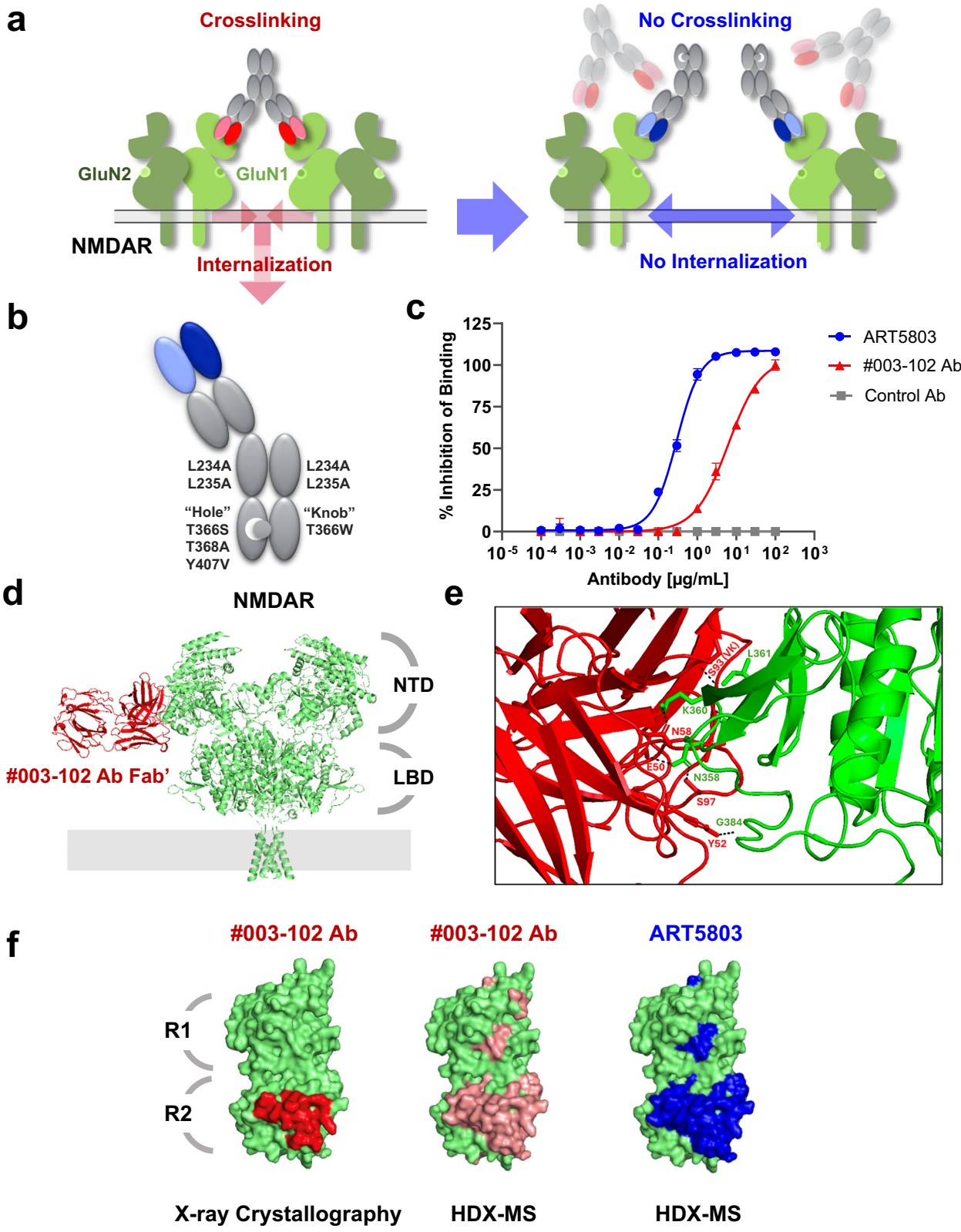

surface of the R2 subdomain. #003-102 Ab interacted with two loops and a short helix of GluN1-NTD via both heavy and light chains. Amino acid residues that were within 4.5Å from #003-102 Ab are listed in Supplemental Table S2. In the tetrameric structure of the NMDAR, the binding surface is located at the side of the receptor, and thus the pathogenic autoantibody is likely to bind adjacent NMDARs with its two arms to crosslink them (Fig. 1d).

The GluN1-NTD is separated from the domain involved in gluta-mate/glycine ligand binding and activation of the receptor, referred to as the ligand binding domain (LBD) (Fig. 1d). The overall structure of the GluN1-NTD bound with #003-102 Ab Fab' was almost identical to that of GluN1-NTD reported in the full NMDAR complex (PDB ID: 8E96, RMSD: 1.156Å), which indicates that no substantial conformational changes are induced by anti-NMDAR autoantibodies binding to this

**Fig. 1 | Generation of ART5803, a humanized monovalent antibody, and its binding characterization to the GluN1-NTD. a** The proposed mechanism of action for the pathogenic autoantibody (bivalent, two-armed, red) and therapeutic antibody (monovalent, one-armed, blue). Bivalent pathogenic anti-NMDAR autoantibodies bind to GluN1-NTD and crosslink adjacent NMDARs, which triggers receptor internalization. Monovalent therapeutic antibodies bind to GluN1-NTD, block binding of pathogenic autoantibodies, and prevent crosslinking and internalization of NMDARs. **b** ART5803 schematic design. Variable regions (antigen binding sites) are shown in blue. "Hole" mutations (T365S / L368A / Y407V) are introduced in the CH3 domain of the parental antibody heavy chain. A "Knob" mutation (T366W) was introduced in the CH3 domain of the second Fc fragment. LALA mutations (L234A / L235A) are introduced in the CH2 domains to avoid antibody-dependent cell-mediated cytotoxicity (ADCC), antibody-dependent cell-mediated phagocytosis (ADCP) or complement-dependent cytotoxicity (CDC). **c** Competitive ELISA between biotinylated #003-102 Ab and ART5803, #003-102 Ab and control Ab on GluN1-NTD. Data are mean ± SD of $n = 2$ internal replicates. Source data are provided as a Source Data file. **d** Epitope mapping by X-ray co-crystallography. Ribbon model of #003-102 Ab (Fab') (red) on NMDAR (green). NTD terminal domain, LBD Ligand Binding Domain. **e** The zoomed-in view of the Fab binding site of #003-102 Ab (red) at the R2 lobe of GluN1-NTD (green) as determined by X-ray crystallography. Interacting residues for #003-102 Ab are from the Vh domain except where indicated with VK. #003-102 Ab residues are labeled with the Kabat numbering system. **f** Space filling model of GluN1-NTD of NMDAR (green) with #003-102 Ab (red) epitope by X-ray crystallography and #003-102 Ab (pink) and ART5803 (blue) epitopes as determined by HDX-MS.

epitope (Supplementary Fig. S3). These results corroborate the findings that binding of some patient's autoantibody Fab fragments to the GluN1 does not have a direct impact on NMDAR activity[11].

Next, we compared putative NMDAR binding epitopes of #003-102 Ab-GluN1-NTD and ART5803-GluN1-NTD complexes using Hydrogen Deuterium Exchange Mass Spectrometry (HDX-MS). Critically, all definitive GluN1-NTD binding sites for #003-102 Ab identified by X-ray co-crystallography were included in putative epitopes identified by HDX-MS. HDX-MS demonstrated that both #003-102 Ab and ART5803 bind to nearly identical epitopes on the GluN1-NTD of the NMDAR, primarily in the R2 subdomain, with partial epitope coverage in the R1 domain (Fig. 1f and Supplementary Table S2).

## ART5803 blocks NMDAR internalization and NMDAR hypofunction driven by pathogenic autoantibody (#003-102 Ab)

We next confirmed the ability of pathogenic #003-102 Ab to reduce NMDAR surface expression by receptor internalization on HEK293 cells expressing human NMDAR GluN1 and GluN2B subunits (NMDAR-expressing HEK293 cells) by a cell-based flow cytometry assay that detects surface NMDAR expression levels (Fig. 2a and Supplementary Fig. S4). HEK293 cells have been widely used for receptor internalization studies, including ion channel receptors[24,25]. While 2 hours (h) of exposure to #003-102 Ab evoked concentration-dependent reduction of NMDAR surface expression, an isotype control antibody did not show any reduction (Fig. 2a, b). In contrast, ART5803 (monovalent) did not show any reduction of NMDAR surface expression (Fig. 2a, b). We compared the activity of ART5803 with a bivalent form of ART5803 (bivalent ART5803) or Fab fragment of ART5803 (Fab ART5803, monovalent) on NMDAR surface expression (Fig. 2c). Neither ART5803 nor Fab ART5803 showed any reduction of NMDAR surface expression. On the contrary, the robust reduction of NMDAR surface expression was observed in response to bivalent ART5803, demonstrating the validity of the monovalent/bivalent model[12]. These data also suggest that NMDAR-expressing HEK293 cells can be useful to detect and quantify NMDAR internalization induced by pathogenic antibodies.

To assess the ability of ART5803 to block NMDAR internalization caused by #003-102 Ab, NMDAR-expressing HEK293 cells were exposed to #003-102 Ab (10 μg/mL) co-incubated with increasing concentrations of ART5803 for 2 h, and then subjected to flow cytometry for NMDAR surface expression. ART5803 blocked the ability of #003-102 Ab to induce NMDAR internalization with an EC$_{50}$ of 0.55 μg/mL (Fig. 2d, Blocking). To assess the ability of ART5803 to rescue NMDAR surface expression after the reduction by pathogenic autoantibodies, #003-102 Ab (10 μg/mL) was incubated with cells for 2 h, followed by co-incubation with increasing concentrations of ART5803 for an additional 2 h. ART5803 rescued NMDAR surface expression with an EC$_{50}$ of 0.69 μg/mL (Fig. 2d, Rescue). As both the blocking and rescue profiles of ART5803 against #003-102 Ab induced NMDAR internalization were indistinguishable, most of the subsequent efficacy studies were conducted testing the ability of ART5803 to block autoantibody pathogenicity.

The monovalent/bivalent model also demonstrated that Fab (monovalent) fragments of some anti-NMDAR antibodies do not impact NMDAR activity directly[11]. To confirm that ART5803 has no impact on NMDAR function, we used NMDAR-expressing HEK293 cells and evaluated Ca$^{2+}$ influx, a well-established NMDAR function, by using the Fluorometric Imaging Plate Reader (FLIPR) assay where intracellular Ca$^{2+}$ is measured using a calcium-sensitive fluorescent dye[26]. As shown in Fig. 2e, incubation of cells with ART5803, a control antibody, or MK-801 (NMDAR antagonist) did not increase intracellular Ca$^{2+}$ levels. Additionally, ART5803 and a control antibody did not decrease NMDA (NMDAR agonist)-induced intracellular Ca$^{2+}$ changes. These results demonstrate that ART5803 has no intrinsic agonistic or antagonistic effects on NMDARs.

To test whether ART5803 can block the reduction in NMDAR-mediated Ca$^{2+}$ influx driven by #003-102 Ab, we again measured Ca$^{2+}$ influx in NMDAR-expressing HEK293 cells by FLIPR assay. In these experiments, cells were incubated with #003-102 Ab or a control IgG1 antibody (1.0 μg/mL each) for 15 minutes (min). ART5803 or the control antibody (0.1-10 μg/mL each) were added to the cells and further co-incubated for 6 h. After incubation, NMDA-induced Ca$^{2+}$ levels were measured. Pathogenic #003-102 Ab significantly reduced Ca$^{2+}$ influx in response to NMDA, demonstrating NMDAR hypofunction (Fig. 2f). ART5803 blocked Ca$^{2+}$ influx reduction induced by #003-102 Ab treatment (Fig. 2f). The blocking efficacy of NMDAR hypofunction was ART5803-concentration dependent, and significant blocking effects were observed at concentrations above 0.1 μg/mL compared to the control Ab.

These experiments demonstrated that blocking of pathogenic #003-102 Ab binding to the GluN1-NTD by ART5803 prevents NMDAR crosslinking, internalization, and hypofunction driven by #003-102 Ab.

## ART5803 rescued NMDAR internalization and spine shrinkage caused by pathogenic autoantibody (#003-102 Ab) in mouse hippocampal neurons

The NMDAR GluN1 subunit is one of the most conserved proteins in vertebrate evolution, with GluN1-NTD sequences showing 100% homology between humans and cynomolgus monkeys, 99.7% between humans and marmosets, 98.5% and 98.2% between humans, rats and mice, respectively. ART5803 and #003-102 Ab demonstrated comparable binding activity to all GluN1-NTD proteins corresponding to humans (cynomolgus monkeys), marmosets, rats and mice GluN1 (Supplemental Fig. S5).

To investigate the consequence of pathogenic autoantibodies and ART5803 on NMDAR internalization and dendritic spine morphology, we used live imaging of surface NMDARs and spine size in cultured hippocampal slices (schematic of experimental design in Fig. 3a). In brief, we imaged surface NMDAR levels using the GluN2A subunit of the NMDAR fused with a pH-sensitive green fluorescent protein, super-ecliptic pHluorin (SEP-GluN2A), which is quenched

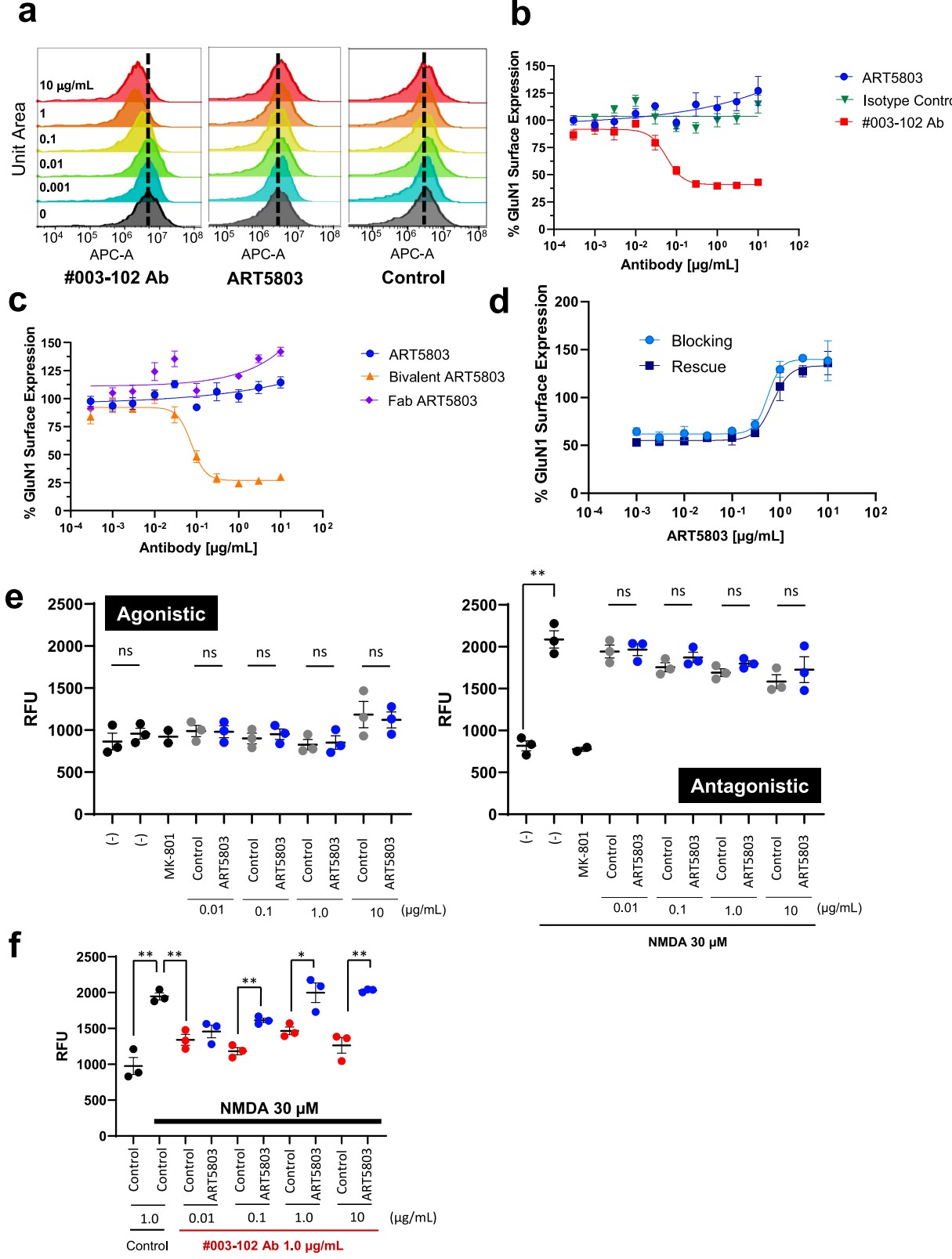

when internalized[27,28]. Neurons were co-transfected with a red fluorescent TdTomato cell fill to monitor spine size. CA1 pyramidal neurons in mouse organotypic slice cultures were co-transfected with TdTomato, SEP-GluN2A, and unlabeled GluN1 at 6-8 days in vitro (DIV) and imaged 7 days later.

In response to #003-102 Ab treatment (25 μg/mL), we observed NMDAR surface expression levels in dendritic spines decreased from

baseline 100% to $72 \pm 4\%$ at 3 h ($p < 0.001$) at 14 h ($p < 0.001$) and spine size decreased from baseline 100% to $84 \pm 6\%$ at 3 h ($p = 0.07$) and $68 \pm 6\%$ at 14 h ($p < 0.001$) (Fig. 3b, e, f; red), but not in response to isotype-matched control antibodies at 25 μg/mL ($105 \pm 5\%$ at 3 h, $110 \pm 7\%$ at 14 h for NMDAR surface expression; $97 \pm 8\%$ at 3 h; $92 \pm 9\%$ at 14 h for spine size, $p > 0.1$) (Supplementary Fig. S6, c, d; black). Importantly, ART5803 alone (25 μg/mL) did not affect NMDAR surface levels or spine

**Fig. 2 | ART5803 blocks NMDAR internalization and NMDAR hypofunction driven by pathogenic autoantibody (#003-102 Ab). a** HEK293 cells expressing human NMDAR GluN1 and GluN2B subunits, were used in an NMDAR internalization assay to assess antibody effects on GluN1 surface expression by flow cytometry. Representative histograms of data used to generate Fig. 2b, show changes in geometric mean fluorescence intensity (GeoMFI) of allophycocyanin fluorescence (APC-A) normalized for visual comparison. Dotted lines indicate 100% GluN1 surface expression. GluN1/GluN2B expressing cells were treated for 2 h with either #003-102 Ab, ART5803 or control antibody (0-10 μg/mL). **b** NMDAR internalization by #003-102 Ab, ART5803 and isotype control at 2 h post-treatment. Data are mean of $n = 4$ independent replicates ± SEM. **c** NMDAR internalization by ART5803, bivalent (two-armed) ART5803 and Fab ART5803 at 2 h post-treatment. Data are mean of $n = 4$ independent replicates ± SEM. **d** ART5803 block (#003-102 Ab + ART5803 co-incubation for 2 h) and rescue (#003-102 Ab pre-treatment for 2 h, ART5803 post treatment for 2 h) of #003-102 Ab induced NMDAR internalization. Data are mean of $n = 3$ independent replicates ± SEM. **e** Assessment of ART5803's agonistic or antagonistic effects on NMDAR was performed using a

Fluorometric Imaging Plate Reader (FLIPR) assay to measure intracellular $Ca^{2+}$ levels with a calcium-sensitive fluorescent dye in HEK293 cells expressing human NMDAR GluN1 and GluN2B subunits. In agonistic activity assessments, ART5803, human IgG isotype control antibody, MK-801, or assay buffer only were added, and fluorescence data was acquired over 5 min. Sequentially for antagonistic activity assessments, NMDA or assay buffer only was added, and fluorescence data was acquired over 5 min. Data are mean ± SEM of $n = 3$ independent replicates and were analyzed by unpaired, two-tailed Student's t-test. ** $p < 0.01$, ns = not significant. RFU relative fluorescence units. **f** Blocking activity of ART5803 on NMDAR hypofunction ($Ca^{2+}$ influx reduction) induced by #003-102 Ab. Cells preincubated with control antibody and subsequently treated with control antibody (conditions on the left side of graph) were either stimulated with assay buffer only or NMDA to demonstrate baseline activity and maximum activation, respectively. Data are mean ± SEM of $n = 3$ independent replicates. Statistical analysis were performed using unpaired, two-tailed, Student's t-tests. * $p < 0.05$, ** $p < 0.01$. RFU relative fluorescence units. Source data is provided as a Source Data file.

size (106 ± 8% at 3 h, 99 ± 7% at 14 h for NMDAR surface expression; 116 ± 14% at 3 h, 110 ± 11% at 14 h for spine size, $p > 0.1$) (Fig. 3c, e, f; blue), demonstrating no adverse or off target effects of ART5803.

We next assessed whether ART5803 treatment could block the effects of #003-102 Ab on NMDAR surface expression and spine size. Notably, administering ART5803 (25 μg/mL) together with #003-102 Ab (25 μg/mL) prevented the decreases in NMDAR surface expression (96 ± 5% at 3 h; 95 ± 6% at 14 h) and spine size (102 ± 4% at 3 h; 101 ± 8% at 14 h) from baseline levels ($p > 0.1$) (Supplementary Fig. S6b, c, d purple), supporting that co-administration of ART5803 blocks NMDAR-internalization induced by #003-102 Ab.

Finally, to assess whether ART5803 treatment could rescue the effects of #003-102 Ab on NMDAR surface expression and spine size, we first treated the slice cultures with #003-102 Ab (25 μg/mL) for 3 h, at which time NMDAR surface expression (80 ± 4%, $p < 0.01$) and spine size (87 ± 4%, $p < 0.05$) were significantly decreased from baseline levels. We then added ART5803 (25 μg/mL) directly after the 3 h imaging time point. Notably, at 14 h, we observed a rescue of both NMDAR surface expression (from 80 ± 4% at 3 h to 92 ± 5%, $p = 0.1$) and spine size (from 87 ± 4% at 3 h to 102 ± 7%, $p < 0.05$) to levels that were not different from baseline levels ($p > 0.1$) (Fig. 3d, e, f; red, then purple).

### ART5803 reversed behavioral abnormalities induced by pathogenic autoantibody (#003-102 Ab) in a marmoset model of anti-NMDAR encephalitis

To test the in vivo efficacy of ART5803, we established an animal model of anti-NMDAR encephalitis using marmosets, a non-human primate.

Previous reports demonstrated that B-cells are able to infiltrate and reside in the CNS of anti-NMDAR encephalitis patients and produce pathogenic anti-NMDAR autoantibodies intrathecally in the CNS compartment at detectable levels in CSF[6,13]. To mimic continuous production of autoantibodies by intracerebral B-cells, we utilized a continuous intracerebroventricular (ICV) infusion of pathogenic #003-102 Ab into the third ventricle with a cannula. Through a preliminary dose range finding study of #003-102 Ab in marmosets, we established an ICV dose of #003-102 Ab (10 μg/h), which reached CSF concentrations around 10 – 20 μg/mL that was required to evoke and maintain robust behavioral abnormalities (Supplementary Table S3). Abnormal behaviors and movements were quantified using a modified version of the Abnormal Rating Scale (ARS), previously used in marmoset Parkinson's disease models[29] (Supplementary Table S4), adapted here to reflect behavior and movement dysfunctions resembling anti-NMDAR encephalitis patients.

First, the efficacy of ART5803 was tested in this marmoset model using the same route (ICV infusion) and dose (10 μg/h) as #003-102 Ab to confirm that the ART5803 blocking effects observed during in vitro and ex vivo studies are translated to in vivo efficacy (Fig. 4a). In our

preliminary study, we confirmed that continuous ICV infusion of ART5803 alone for 2 weeks was well tolerated and did not evoke noticeable behavioral abnormalities in marmosets. Cannulas were implanted into the third ventricle of marmoset brains, and 10 μg/h of #003-102 Ab was continuously infused with micro-infusion pumps. Two weeks after continuous ICV infusion of #003-102 Ab (Day 14), marmosets exhibited robust behavioral abnormalities with a significant increase in ARS scores compared to baseline before pathogenic #003-102 Ab infusion (1.8 ± 0.4 at baseline to 9.6 ± 2.1 after 14 days infusion, $p < 0.01$) (Fig. 4b, c). From Day 14, continuous ICV infusion of ART5803 or control antibody (10 μg/h) was administered alongside the ICV infusion of #003-102 Ab for an additional 2 weeks. Co-administration of ART5803 for 2 weeks significantly reduced abnormal behaviors with ARS scores decreasing from 9.6 ± 2.1 at Day 14 to 3.6 ± 1.0 at Day 28 ($p < 0.01$) (Fig. 4b, c, Supplementary Table S5 and Supplementary Movies 1–3). Control antibody treatment was ineffective with ARS scores remaining similar, from 8.0 ± 4.4 at Day 14 to 9.0 ± 1.0 at Day 28 ($p > 0.05$).

Next, we investigated whether ART5803 delivered via peripheral administration (intraperitoneal, IP) could reverse abnormal behaviors in the marmoset model (Fig. 5a). As a general assumption, monoclonal antibody penetration ratio into the brain (CSF) from blood is low (0.1 – 0.2%)[30]. CSF concentrations of #003-102 Ab by ICV infusion and ART5803 by IP injections in marmosets were assessed to determine the optimal study design (Supplementary Fig. S7). The steady state CSF concentration of #003-102 Ab by ICV continuous infusion (10 μg/h) was approximately 10 – 20 μg/mL. As NMDAR internalization induced by #003-102 Ab at 10 – 20 μg/mL was blocked by ART5803 at 1 – 2 μg/mL in NMDAR-expressing HEK cells, we predicted that 1–2 μg/mL of ART5803 in the CSF would be necessary to block #003-102 Ab pathogenicity in this marmoset model. The IP dose, 400 mg/kg twice a week, to achieve a target concentration of 1–2 μg/mL of ART5803 in the CSF was determined by a preliminary pharmacokinetics (PK) study and simulation in marmosets (Supplementary Fig. S7). Through PK studies, we confirmed that IP injections of ART5803 alone at doses of 200, 400, and 800 mg/kg twice a week for 2 weeks were well-tolerated and did not induce any noticeable behavioral abnormalities in marmosets (Supplementary Table S6).

#003-102 Ab (10 μg/h) was continuously ICV infused for 3 weeks to evoke and maintain behavior and motor abnormalities in marmosets. Six days after ICV infusion of #003-102 Ab (Day 6), marmosets showing robust abnormalities were divided into two treatment groups, ART5803 ($n = 8$) and control vehicle ($n = 7$) and dosed with either 400 mg/kg ART5803 or vehicle via IP twice a week for 2 weeks starting on Day 7 (Fig. 5a). ICV administration of #003-102 Ab for 6 days resulted in robust abnormal behaviors in both treatment groups, with ARS scores increasing from 2.4 ± 0.5 at baseline (Day 0) to 11.4 ± 0.8

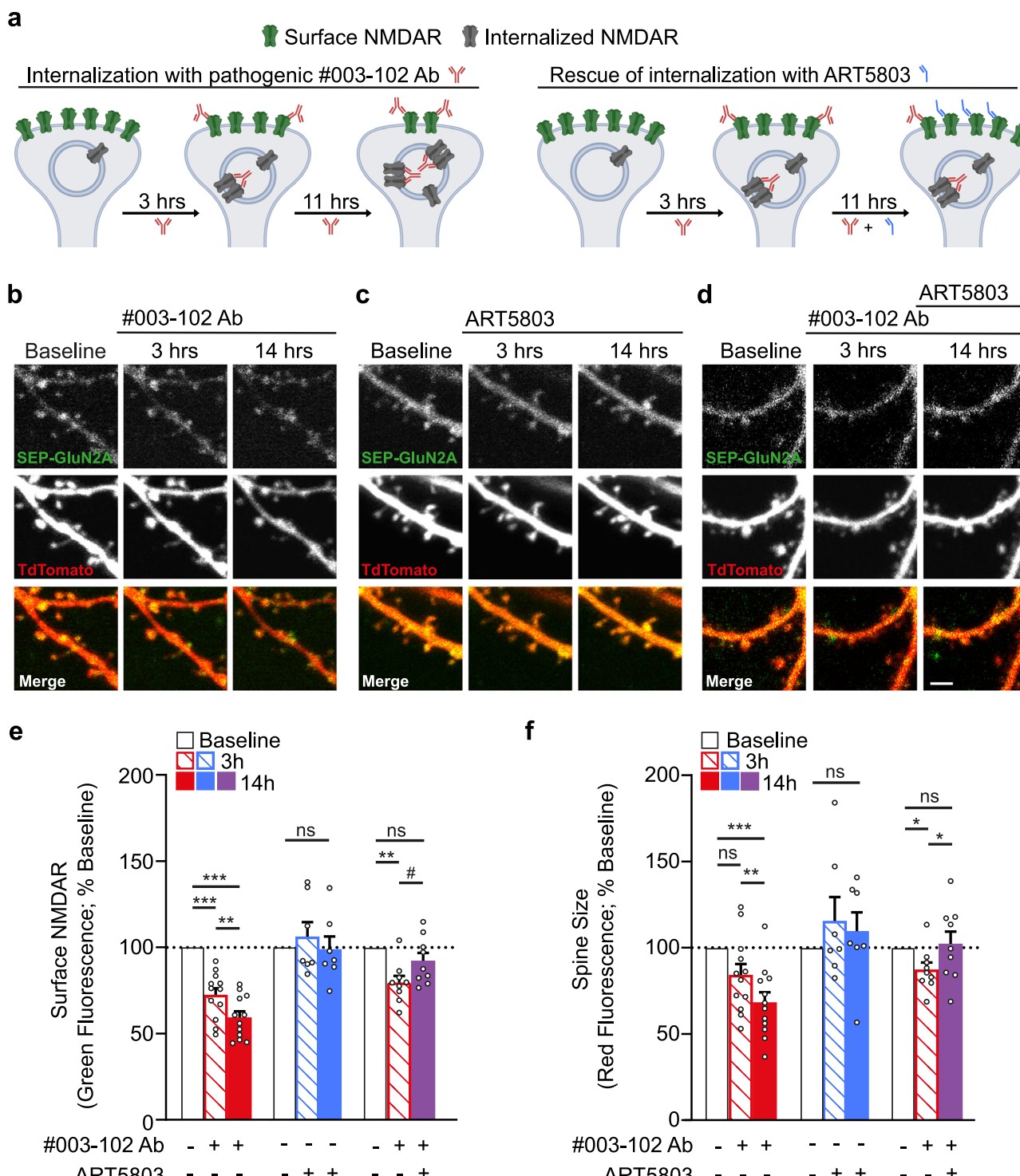

**Fig. 3 | ART5803 rescued NMDAR internalization and spine shrinkage caused by pathogenic autoantibody (#003-102 Ab) in mouse hippocampal neurons. a** Schematic of experimental paradigm. Super-ecliptic pHluorin (SEP)-tagged NMDAR is fluorescent at the surface (green) but is quenched when internalized (gray). Created in BioRender[69]. **b–d** Images of dendrites from CA1 pyramidal neurons in hippocampal slice cultures transfected with SEP-GluN2A (green), GluN1 (unlabeled), and TdTomato (red) directly prior to (baseline) and after (3 h and 14 h) incubation with (**b**) pathogenic #003-102 Ab, or (**c**) ART5803, or (**d**) pathogenic #003-102 Ab alone followed by ART5803 at 3 h. The scale bar is 2 μm. **e, f** Reduction of (**e**) surface NMDAR expression and (**f**) spine size driven by pathogenic #003-102 Ab (left, red; 12 cells) was rescued by the addition of ART5803 at 3 h (right, red and purple; 9 cells). ART5803 alone (middle, blue; 7 cells) had no impact on surface NMDAR expression or spine size. Each point is from an individual spine, and bars represent mean ± SEM. Paired ordinary one-way ANOVA with Tukey's multiple comparisons test. *p < 0.05; **p < 0.01, ***p < 0.001, #p = 0.1, ns = not significant. Source data are provided as a Source Data file.

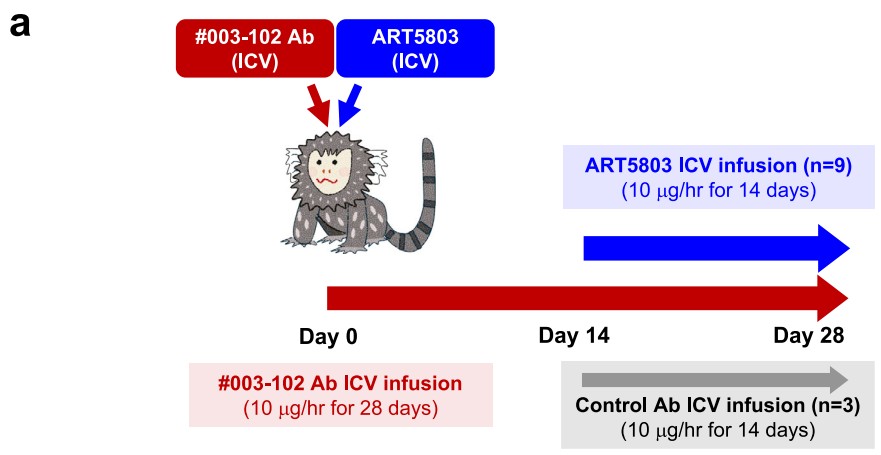

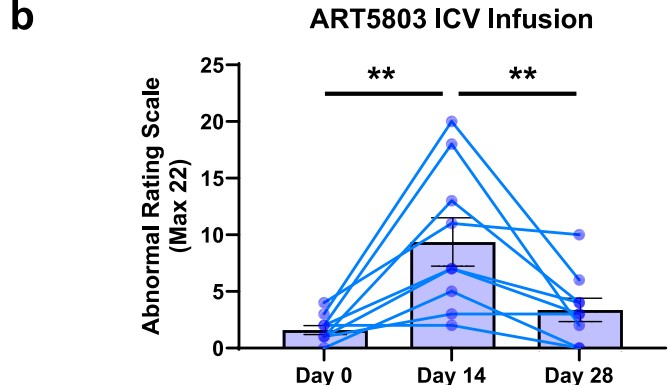

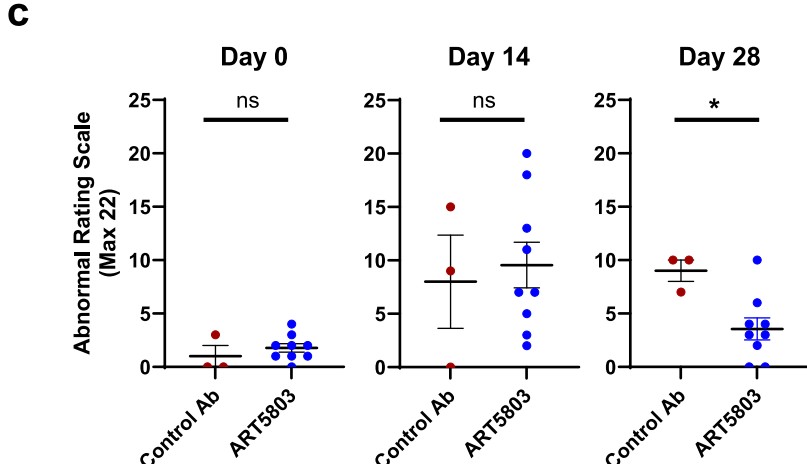

**Fig. 4 | ICV infusion of ART5803 reverses abnormal behaviors induced by pathogenic autoantibody (#003-102 Ab) in marmoset disease model.**
**a** Experimental design of ART5803 intracerebroventricular (ICV) infusion efficacy study. Continuous ICV infusion of pathogenic #003-102 Ab (10 μg/h) into the third ventricle in marmosets for 4 weeks. Two weeks after ICV infusion of #003-102 Ab, continuous ICV infusion of ART5803 ($n = 9$) or control antibody ($n = 3$) at 10 μg/h was administered alongside the ICV infusion of #003-102 Ab for an additional 2 weeks. Over the course of the study, abnormal behaviors were evaluated using the abnormal rating scale. **b** Time course of abnormal behaviors (Abnormal Rating Scale, maximum score 22) in marmosets with ART5803 ICV infusion at Day 0, Day

14, and Day 28. Each point is from an individual marmoset ($n = 9$) and bars represent mean ± SEM. Statistical analysis was performed between timepoints using two-tailed, Wilcoxon matched-pairs signed rank test. **, $p < 0.01$. **c** Comparison of abnormal behaviors between ART5803 and control antibody ICV-infused groups at Day 0, Day 14 and Day 28 of the study. Each point is from an individual marmoset and data are mean ± SEM of $n = 3$ (#003-102 Ab + Control antibody) or $n = 9$ (#003-102 Ab + ART5803). Statistical analysis was performed using unpaired, two-tailed, Student's t-tests. *, $p < 0.05$; ns = not significant. Source data are provided in Supplementary Table S5.

(Day 6) ($p < 0.01$) for the ART5803 treatment group and 2.0 ± 0.7 to 12.1 ± 0.8 ($p < 0.01$) for the vehicle treatment group. After 1 week post ART5803 400 mg/kg IP injections, the treatment group showed a significant reduction in ARS scores from 11.4 ± 0.8 at Day 6 to 7.3 ± 0.8 at Day 14 ($p < 0.01$). ARS scores of the ART5803 treatment group 7.3 ± 0.8

were significantly lower compared to the vehicle treatment group 11.4 ± 1.4 at Day 14 ($p < 0.05$). This benefit was maintained for 2 weeks with a significant reduction in ARS scores from 11.4 ± 0.8 at Day 6 to 7.5 ± 0.8 at Day 21 ($p < 0.05$), and significantly lower compared to the vehicle treatment group 10.6 ± 1.2 at Day 21 ($p < 0.05$) (Fig. 5b, c,

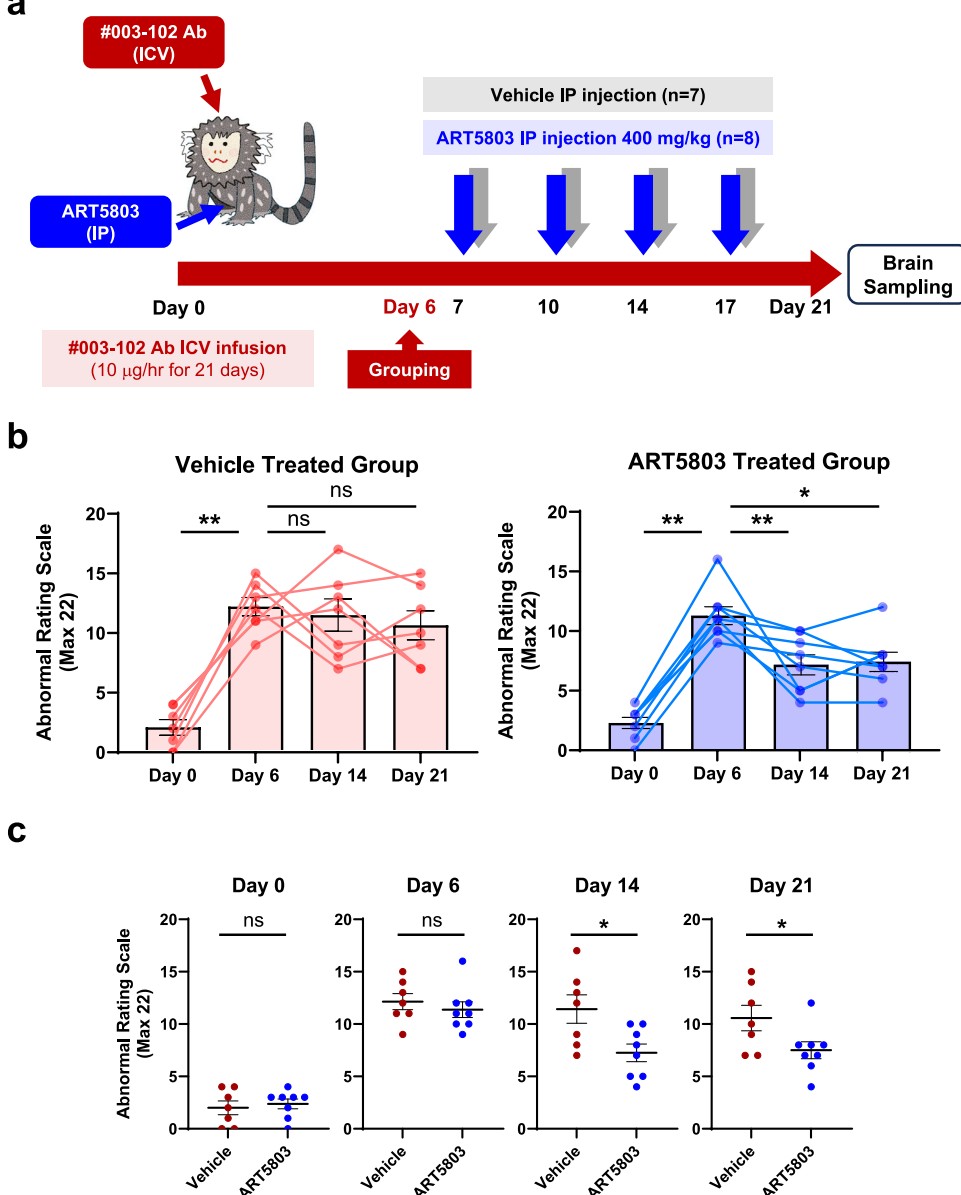

**Fig. 5 | IP injections of ART5803 reverse abnormal behaviors induced by pathogenic autoantibody (#003-102 Ab) in marmoset disease model.**
**a** Experimental design of ART5803 intraperitoneal (IP) injection efficacy study. Continuous ICV infusion of pathogenic #003-102 Ab (10 μg/h) into the third ventricle in marmosets for 3 weeks. Six days after ICV infusion of #003-102 Ab, marmosets showing robust abnormalities were divided into two treatment groups (Grouping on Day 6), ART5803 ($n = 8$) and control vehicle ($n = 7$) and dosed with either 400 mg/kg ART5803 or vehicle via IP twice a week for 2 weeks starting on Day 7. Over the course of the study, abnormal behaviors were evaluated using the abnormal rating scale. **b** Time course (days) of abnormal behaviors in marmosets

with vehicle or ART5803 IP injections at Day 0, Day 6, Day 14 and Day 21. Each point is from an individual marmoset in the vehicle treated group ($n = 7$) and ART5803 treated group ($n = 8$), and bars represent mean ± SEM. Statistical analysis was performed using two-tailed, Wilcoxon matched-pairs signed rank test. *, $p < 0.05$; **, $p < 0.01$; ns = not significant. **c** Comparison of abnormal behaviors between ART5803 and vehicle IP injected groups at Day 0, Day 6, Day 14 and Day 21 of the study. Each point is from an individual marmoset and data are mean ± SEM of $n = 7$ (Pathogenic antibody + Vehicle) or $n = 8$ (Pathogenic antibody + ART5803). Statistical analysis was performed using unpaired, two-tailed, Student's t-tests. *, $p < 0.05$; ns = not significant. Source data are provided in Supplementary Table S7.

Supplementary Table S7 and Supplementary Movies 4–6). IP injections of ART5803 at 400 mg/kg twice a week for 2 weeks were well-tolerated and demonstrated a potential peripheral route of therapeutic administration in patients.

**Systemic administration of ART5803 restored GluN1 protein expression, reduced by pathogenic autoantibody (#003-102 Ab) in a marmoset model of anti-NMDAR encephalitis**
Previous studies reported robust GluN1/NMDAR expression reduction by pathogenic autoantibodies in rat brains and in anti-NMDAR encephalitis patient brains[12,31]. In order to assess expression levels of GluN1

in marmoset brains of the IP administration cohort (vehicle versus ART5803), after the final ARS scoring (Day 21), the left hemisphere was processed for immunostaining and the cerebral cortex (whole) from the right hemisphere was dissected for western blot analysis. Immunostaining and western blot analysis of GluN1 and GluR2 (AMPA receptor) were performed using well-characterized antibodies[12,32,33].

Immunostaining of marmoset brains from preliminary studies (independent assessment of #003-102 Ab ICV and ART5803 IP) demonstrated different distribution patterns between #003-102 Ab by ICV infusion alone (10 μg/h for 4 weeks) and ART5803 by IP

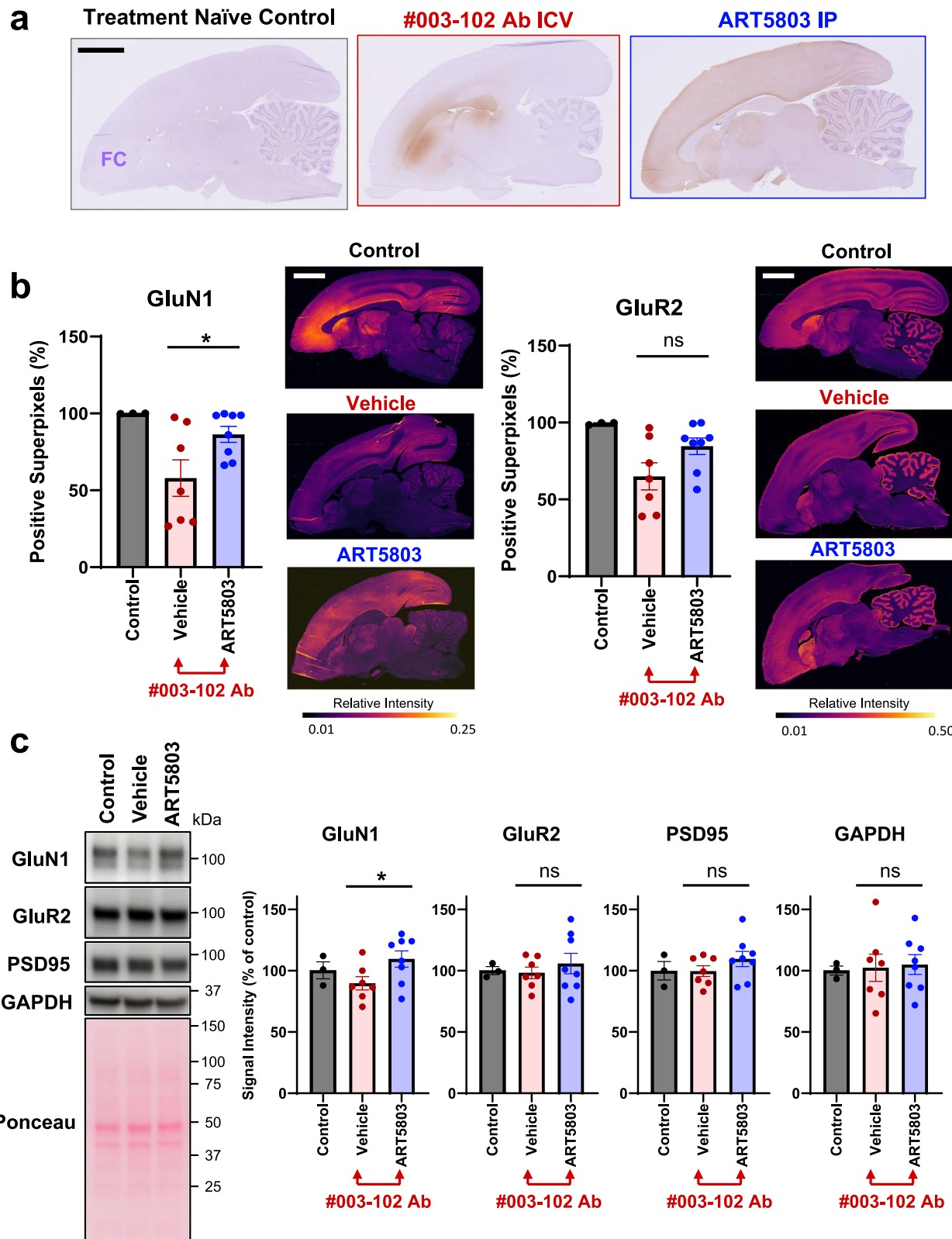

injections alone (400 mg/kg, twice a week for two weeks). While ART5803 distribution was nearly uniform across the brain parenchyma with virtually all neuronal processes in the gray matter bound with ART5803 by IP injections, dense #003-102 Ab staining was noted in the brain areas close to ventricles by ICV infusion (Fig. 6a). In patients, the most prominent decrease in NMDAR expression was reported in the cerebral cortex[31]. Further, denser signals of #003-102 Ab were detected in rostral brain regions due to proximity to ventricles. We therefore completed immunostaining and 3,3′-diaminobenzidine (DAB) signal intensity quantification from the frontal cortex by using QuPath software[34] scripts adapted to brain staining analysis based on the previous literature[35]. GluN1 staining was significantly lower in the #003-102 Ab ICV infusion plus vehicle IP treatment group ($n = 7$) compared to the

**Fig. 6 | IP injections of ART5803 reverse GluN1 reduction induced by pathogenic autoantibody (#003-102 Ab) in marmoset disease model.**
**a** Representative immunostaining patterns of human IgG, with anti-human IgG, 3,3'-diaminobenzidine (DAB) staining as brown, and nuclei hematoxylin staining as blue, in sagittal brain sections of marmosets without treatment (treatment naïve control) or treated with either #003-102 Ab ICV infusion at 10 μg/h for 4 weeks (#003-102 Ab ICV) or ART5803 IP injections at 400 mg/kg twice a week for 2 weeks (ART5803 IP). The treatment naïve control marmoset brain serves as negative background control. Slides were scanned at 40x magnification. The scale bar is 5 mm. FC = frontal cortex. **b** Comparison of GluN1 (NMDAR) and GluR2 (AMPA receptor) levels by immunostaining in the frontal cortex of marmosets without treatment (treatment naïve control, n = 3) or treated with #003-102 Ab ICV plus vehicle IP injections (n = 7) or ART5803 IP injections (n = 8), quantified using DAB positive superpixel segmentation in QuPath. Data are represented as mean ± SEM.

Statistical analysis was performed using unpaired, two-tailed, Student's t-tests. *p < 0.05; ns = not significant. Representative heat maps demonstrating DAB intensity, with black indicating low signal, and yellow, high signal. Scale bars are 5 mm. Source data are provided as a Source Data file. **c** Comparison of GluN1 (NMDAR), GluR2 (AMPA receptor), PSD95 (synapse marker) and GAPDH levels by western blot analysis in the cerebral cortex (whole) of marmosets without treatment (treatment naïve control, n = 3) or treated with #003-102 Ab ICV plus vehicle IP injections (n = 7) or ART5803 IP injections (n = 8). Quantitative comparisons between samples on different blots were normalized to the treatment naïve control for total protein. All samples were from the same experiment and blots were processed in parallel. Data are represented as mean ± SEM. Statistical analysis was performed using two-tailed, unpaired Student's t-tests. *p < 0.05; ns = not significant. Uncropped blots are reported in Supplementary Fig. S10.

ART5803 IP treated group (n = 8) (p < 0.05) (Fig. 6b and Supplementary Fig. S8). GluN1 expression recovery by ART5803 was closer to treatment naïve controls. Although GluR2 (AMPA receptor) staining did not show a significant difference between vehicle and ART5803 treated groups, a similar trend showing lower GluR2 expressions in the vehicle treated group and a recovery in the ART5803 treated group was observed (Fig. 6b and Supplementary Fig. S9), which aligns recent findings showing anti-NMDAR autoantibodies reduced AMPA receptor levels presumably due to disruption of NMDAR-regulated synaptic plasticity[36].

We assessed expression levels of GluN1, GluR2, PSD95 (synaptic marker) and GAPDH by western blot analysis in total proteins from whole cerebral cortices (we could not precisely dissect frontal cortices due to the frozen storage conditions). GluN1 expression levels in the whole cerebral cortex were significantly lower in (#003-102 Ab ICV infusion plus) vehicle IP treatment group (n = 7) compared to ART5803 IP treated group (n = 8) (p < 0.05). Levels of GluR2, PSD95 and GAPDH were not different between vehicle and ART5803-treated groups (Fig. 6c and Supplementary Fig. S10).

### ART5803 blocks NMDAR internalization driven by mixtures of pathogenic monoclonal autoantibodies

To investigate whether ART5803 can block the pathogenicity of autoantibodies in a polyclonal setting, we used mixtures of representative patient-derived monoclonal autoantibodies based on the report by Ly et al.[14]. A mixture of four monoclonal autoantibodies from three patients were tested and the concentrations of each antibody in the mixture were adapted from the report to reflect concentrations observed in a patient's CSF. NMDAR-expressing HEK293 cells were co-incubated with the pathogenic autoantibody mixture for 18 h, then ART5803 was added to the cells and further incubated for 30 h. ART5803 blocked NMDAR internalization induced by pathogenic polyclonal antibodies in a concentration-dependent manner (Fig. 7a). ART5803 could rescue almost 100% of NMDAR expression at 1.0 μg/mL. The control antibody had no effect on autoantibody-induced NMDAR internalization.

Furthermore, we tested ART5803's blocking activity against five different monoclonal autoantibodies mixtures to represent five patients' CSF conditions based on the report by Ly et al.[14] Monoclonal autoantibody mixtures reflecting CSF conditions of patients #2, #3 and #4, which contained high concentrations of #003-102 Ab (#003-102) and #008-218 (the second strongest in Ly et al. 2018), attenuated NMDAR cell surface expression by about 80%. On the other hand, monoclonal autoantibody mixtures from patients #5 and #6 showed approximately a 50% reduction in NMDAR expression, despite containing higher amounts of autoantibodies compared with patients #2 - #4. ART5803 blocked NMDAR internalization induced by all patient-derived autoantibody mixtures, achieving nearly complete blocking at 1.0 μg/mL (Fig. 7b).

### ART5803 blocks NMDAR internalization driven by anti-NMDAR encephalitis patients' sera and CSF

To further test the effectiveness and generality of ART5803, we obtained and investigated sera and CSF from multiple anti-NMDAR encephalitis patients. To compare binding activity and pathogenicity (NMDAR internalization) of each patient sample, we generated 502 Ab, a recombinant pathogenic antibody with a strong affinity to NMDAR and in vitro pathogenicity ($EC_{50}$ = 0.021 μg/mL, see *Methods*). To determine binding and pathogenic activity of patients' sera and CSF, 502 Ab was used as a standard pathogenic antibody for quantitation.

First, sera and CSF from seven patients were evaluated for NMDAR binding by a cell-based flow cytometry assay using NMDAR-expressing HEK293 cells to obtain pseudo-concentration equivalents to 502 Ab binding activity. Pseudo-concentrations of anti-NMDAR autoantibodies were determined for six patients' serum and all patients' CSF based on 502 Ab binding to NMDAR. Autoantibody concentrations ranged from 0.24 to 2.2 μg/mL (502 Ab equivalent) in serum and were lower in CSF ranging from 0.0078 to 0.15 μg/mL (502 Ab equivalent). These concentrations showed a significant correlation between serum and CSF levels ($R^2$ = 0.847, p = 0.0093, n = 6) (Fig. 7c and Table 1). Then, the pathogenicity of seven patients' sera and CSF was evaluated by the NMDAR internalization assay in comparison to the reference antibody 502 Ab. All patients' sera showed NMDAR internalization activity ranging from 1.0 to 5.5 μg/mL equivalent to 502 Ab internalization activity. Due to the limited volume of CSF samples available and low sensitivity of the NMDAR internalization assay, we detected robust reduction of NMDAR surface expression from only three CSF samples. These samples induced detectable NMDAR internalization activity ranging from 0.56 to 1.4 μg/mL (502 Ab equivalent).

ART5803 blocked NMDAR internalization induced by sera and CSF from patients in a concentration-dependent manner. The effective concentration of ART5803 for patient serum-induced NMDAR internalization blocking was determined to be 3.0–10 μg/mL. The concentration of ART5803 required to completely block the NMDAR internalization caused by CSF from patients #2 and #3 was 1.3 μg/mL and 0.63 μg/mL respectively. Although the NMDAR internalization caused by CSF from patient #1 was not completely blocked, ART5803 at 2.5 μg/mL was able to maintain the NMDAR surface expression level at 74%, which is roughly double the level from 42% reduced by strong patient #1 CSF (Fig. 7d and Table 2).

### Cynomolgus monkey population PK (PopPK) modeling with human scaling and simulation demonstrated ART5803 IV dosing feasibility in patients

A study was conducted in cynomolgus monkeys to determine the PK profile and CSF penetration of ART5803. The ART5803 PK profile in cynomolgus monkeys was similar to that typically seen with most monoclonal antibodies with a human IgG1 backbone.

In cynomolgus monkeys, a single 30 min IV infusion of ART5803 at 100 mg/kg and 1000 mg/kg resulted in dose

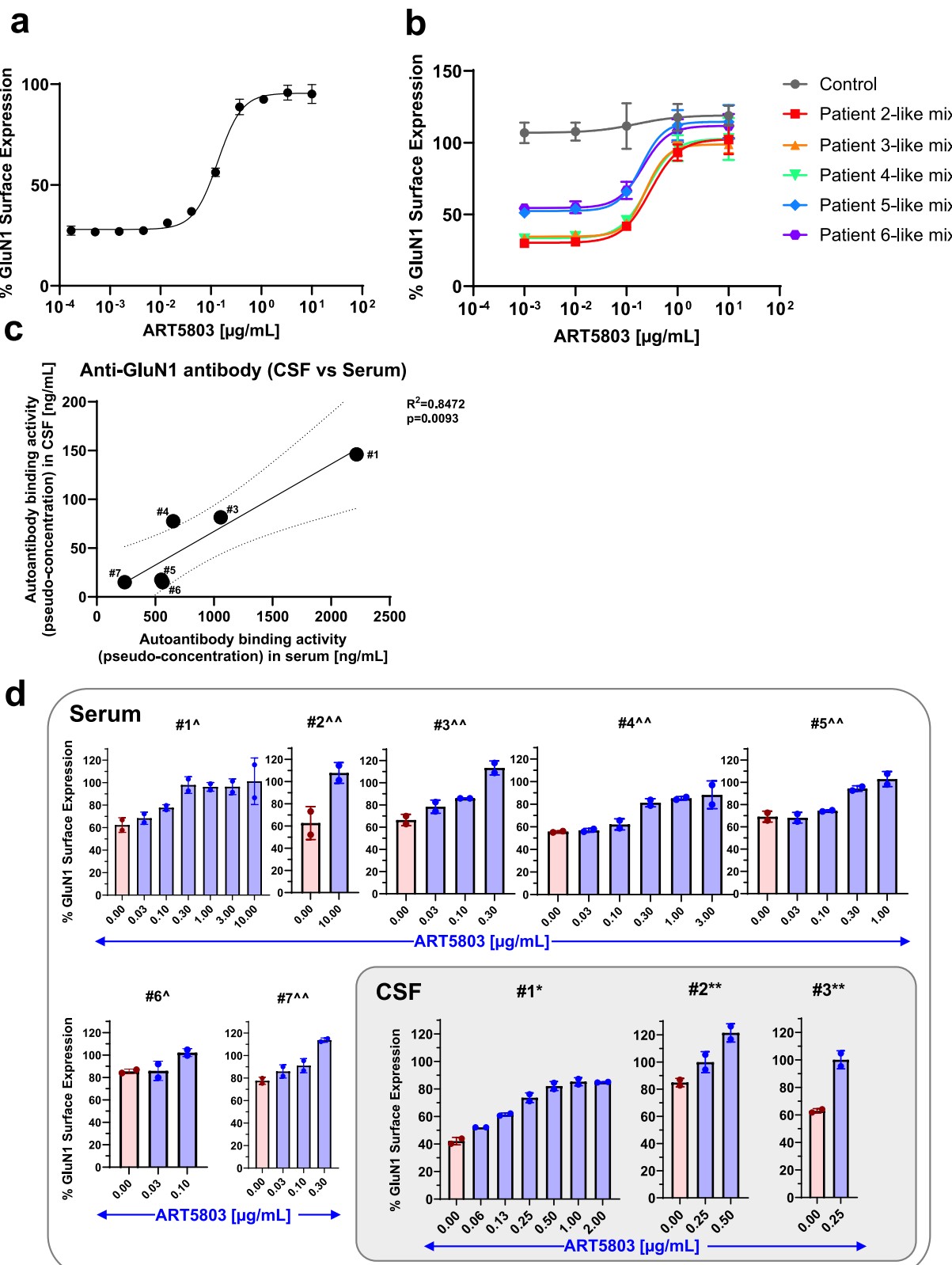

proportional maximum concentration ($C_{max}$) and area under the curve (AUC) exposure both in the serum and CSF (Fig. 8a and Supplementary Table S8a,b). The time to maximal concentration ranged from 5 to 30 min in the serum ($C_{max}$ 2400 ± 340 µg/mL at 100 mg/kg and 21000 ± 4300 µg/mL at 1000 mg/kg) and 24 and 168 h in the CSF ($C_{max}$ 3.0 ± 2.4 µg/mL at 100 mg/kg and 15 ± 8.0 µg/mL at 1000 mg/kg). The mean half-lives were 237 h and

220 h in the serum, and 208 h and 339 h in the CSF at a dose of 100 mg/kg and 1000 mg/kg, respectively. Overall, the concentration observed in the CSF was significantly lower than that measured in the serum at all comparable timepoints. The exposure of ART5803 in the CSF as compared to serum, measured by $C_{max}$ and AUC (0-1008 h), was 0.13% and 0.26% at a dose of 100 mg/kg and 0.072% and 0.19% at a dose of 1000 mg/kg, respectively.

**Fig. 7 | ART5803 blocks NMDAR internalization induced by patient-derived polyclonal autoantibodies and patients' sera and CSF in NMDAR-expressing HEK293 cells. a** Using HEK293 cells expressing human NMDAR GluN1 and GluN2B subunits, the NMDAR internalization assay evaluated the effects of antibodies on GluN1 surface expression, measured by flow cytometry. Recovery of surface GluN1 expression after incubation with a patient monoclonal antibody mixture for 18 h, followed by ART5803 co-incubation for an additional 12 h. Data are mean ± SD of $n = 2$ internal replicates. **b** Effect of ART5803 (blocking, co-incubation) on Patient 2-6 CSF-like monoclonal antibody mixtures induced internalization for 24 h. Data are mean ± SD of $n = 2$ internal replicates. **c** NMDAR binding activity of patients' sera and CSF samples to HEK293 cells expressing human NMDAR GluN1 and GluN2B subunits, measured by flow cytometry (502 Ab equivalent pseudo-concentration).

Each ID correlates to a sera/CSF matched patient. $R^2$ and $p$ values were calculated by simple linear regression ($n = 6$; data presented as mean). Simple linear regression best fit line is solid, and the 95% confidence bands are dotted. Patient #2 was excluded from this graph due to the lack of serum data. CSF = cerebrospinal fluid. **d** The effect of ART5803 (blocking, co-incubation) on NMDAR internalization induced by patients' sera and CSF for 20 h. Each graph is titled with a unique patient identifier and is matched between serum and CSF. Serum dilutions of 3% and 10% are indicated by ^ and ^^ respectively. CSF dilutions of 10% and 40% are indicated by * and ** respectively. Red bars indicate no treatment, and blue bars indicate treatment with ART5803. The mean ± SD of $n = 2$ internal replicates is represented for all samples except for serum #1, which the mean ± SD from $n = 2$ independent experiments. Source data are provided as a Source Data file.

Based on ART5803 concentrations needed to block NMDAR internalization induced by patient's CSF and monoclonal antibodies (with concentrations reflecting patients' CSF), we predict that around 0.6–2.5 μg/mL of ART5803 in the CSF would be the target concentration necessary to treat anti-NMDAR encephalitis patients. Cynomolgus monkey PopPK modeling with human scaling and simulation predicted that 40–100 mg/kg IV weekly dosing of ART5803 would achieve the CSF target concentration in patients (Fig. 8b).

## Discussion

The present study demonstrates a therapeutic potential for the monovalent humanized IgG ART5803 as a fast-acting, efficacious, and safe treatment option for patients with anti-NMDAR encephalitis.

We identified the epitope of #003-102 Ab, which is one of the highest affinity and most pathogenic anti-NMDAR autoantibodies cloned from anti-NMDAR encephalitis patients to date[13]. Although epitopes of autoantibodies on the GluN1 subunit have been proposed by mutagenesis studies[15], they have not yet been confirmed by crystallographic techniques. The N368/G369 region on the GluN1-NTD was found to be critical for binding of pathogenic autoantibodies. We determined the epitopes in more definitive ways by X-ray crystallography and HDX-MS and discovered that #003-102 Ab binds to the conformational epitope in the GluN1-NTD (R2 subdomain). This epitope includes amino acid sequences surrounding N368/G369 but does not directly involve these amino acids residues. Findings from this study, together with the previous mutagenesis studies, suggest that the N368/G369 region may be critical for maintaining the native conformation of the GluN1-NTD.

Anti-NMDAR encephalitis is the most prevalent form of autoimmune encephalitis[1] and anti-NMDAR autoantibodies have been widely found in sera of patients with non-encephalitis diseases, such as psychosis[37,38], dementia[39,40], and even in healthy individuals[41].

Importantly, an unmutated germline-configured autoantibody binding to the GluN1-NTD has been found in an anti-NMDAR encephalitis patient[42]. These findings indicate that autoantibodies having a binding affinity to GluN1-NTD exist as a part of the human naïve B cell repertoire and suggest a possibility that these unmutated germline-configured antibodies are part of innate-like components of the immune system that target invading pathogens. In the present study, we have discovered the conformational epitope of #003-102 Ab in GluN1-NTD with discontinuous stretches of amino acids (Supplementary Table S2). Remarkably, the centered amino acid stretch Q357-V362 with L356 (LQNRKLV) is a 100% match with a portion of a Toxoplasma protein in addition to proteins in bacteria that are associated with CNS disorders, but does not completely match with any other proteins beside the NMDAR GluN1 subunit in mammals (Supplementary Table S9). Toxoplasmosis and bacterial infections are well-established risk factors for neuropsychiatric diseases[43–45]. As reported recently in multiple sclerosis[46], molecular mimicry mechanisms may be involved in anti-NMDAR autoantibody generation in humans and other mammals, such as the polar bear[47]. Despite the different origins of #003-102 Ab (anti-NMDAR encephalitis patient) and ART5803 (GluN1-NTD immunized mouse), both antibodies bind to nearly identical epitopes on GluN1-NTD (Fig. 1f and Supplementary Table S2), suggesting the unique and strong immunogenicity of GluN1-NTD epitopes centered by Q357-V362 in mammals.

In the present study, we established a passive-transfer marmoset model for anti-NMDAR encephalitis to test ART5803 efficacy in vivo. Adult marmosets that received continuous ICV infusion of #003-102 Ab developed robust behavior and motor behavioral abnormalities resembling those observed in anti-NMDAR encephalitis patients. As primates, marmosets possess well-developed cerebral cortices, which

### Table 1 | Patient serum and CSF autoantibody NMDAR binding activity equivalent to activity of 502 Ab

| Patient ID | Equivalent binding activity as 502 Ab [μg/mL] | | CSF/Serum Ratio |
|---|---|---|---|
| | Serum | CSF | |
| #1 | 2.2 | 0.15 | 6.8% |
| #2 | ND | 0.0078 | ND |
| #3 | 1.1 | 0.082 | 7.5% |
| #4 | 0.65 | 0.078 | 12.0% |
| #5 | 0.55 | 0.018 | 3.3% |
| #6 | 0.56 | 0.015 | 2.7% |
| #7 | 0.24 | 0.015 | 6.3% |

*ND* Not determined due to availability of sample.
#s indicate unique patient identification numbers.

### Table 2 | Patient serum and CSF autoantibody NMDAR internalization activity equivalent to activity of 502 Ab and minimum effective reversal concentration of ART5803

| Patient ID | Serum | | CSF | |
|---|---|---|---|---|
| | Equivalent to internalization activity of 502 Ab [μg/mL] | Minimum effective conc. of ART5803 [μg/mL] | Equivalent to internalization activity of 502 Ab [μg/mL] | Minimum effective conc. of ART5803 [μg/mL] |
| #1 | 5.5 | 10 | 1.4 | 2.5 (74%) |
| #2 | 1.7 | 10 | 0.56 | 1.3 |
| #3 | 1.5 | 3.0 | 0.94 | 0.63 |
| #4 | 2.0 | 3.0 (81%) | ND | - |
| #5 | 1.4 | 10 | ND | - |
| #6 | 2.6 | 3.3 | ND | - |
| #7 | 1.0 | 3.0 | ND | - |

Values in parentheses indicate the percentage of GluN1 surface expression achieved by treatment with the corresponding concentration of ART5803.

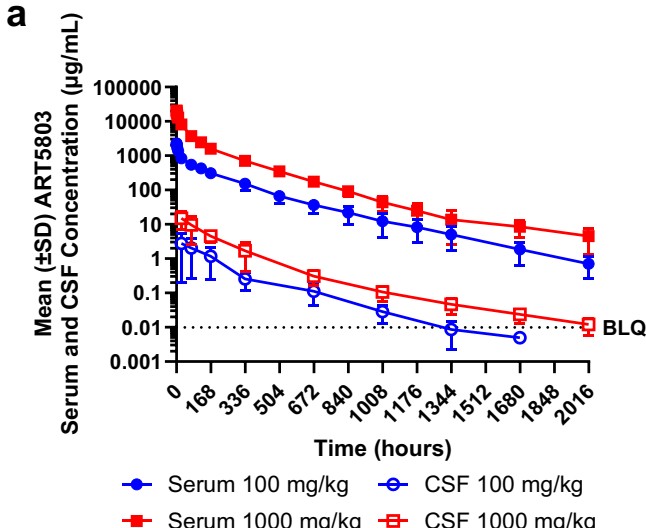

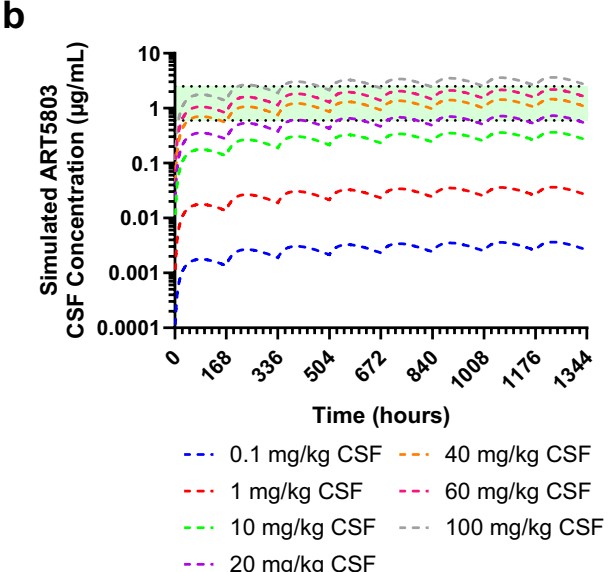

**Fig. 8 | Cynomolgus monkey PK and population PK (PopPK) modeling with human scaling and simulation. a** Mean (± SD) ART5803 overlay of serum and CSF concentrations in cynomolgus monkeys following a single IV dose at 100 and 1000 mg/kg (*n* = 3 each dose). SD = standard deviation, CSF = cerebrospinal fluid, BLQ = below limit of quantification. **b** Simulated ART5803 concentrations in human CSF via IV weekly dosing based on cynomolgus monkey PopPK modeling. The model-predicted CSF concentration–time profiles after IV administration are depicted (0.1 – 100 mg/kg every 7 days). Approximate doses greater than 40, 60 and 100 mg/kg were required to achieve target concentration 0.6 – 2.5 µg/mL in CSF (the top light green band). Source data are provided as a Source Data file.

are the main target regions of anti-NMDAR autoantibodies, resulting in reduced NMDAR densities[31,48].

Despite significant effort, we were unable to establish a mouse model for anti-NMDAR encephalitis. ICV infusion of the pathogenic #003-102 Ab into mice did not elicit observable behavioral changes. Similar results have been reported in a previous study where passive transfer of patient pathogenic antibodies into mice did not result in memory deficits, anxiety-related behaviors, or motor impairment[49], although other studies demonstrated that the passive transfer mouse model showed memory deficits, anhedonic/depressive-like behaviors,

without affecting other behavioral or locomotor tasks[50]. An active immunization mouse model of anti-NMDAR encephalitis has been reported[51], but this model included autoantibodies that were not restricted to the GluN1 subunit of NMDAR and showed severe immune and inflammatory reactions that differed from those seen in the anti-NMDAR encephalitis patients[6]. Previous reports demonstrated species differences in response to hypo-NMDAR status, such as ketamine sensitivity, where mice showed higher tolerability (lower sensitivity) to hypo-NMDAR status compared to primates, including humans[52,53]. These species differences to hypo-NMDAR status prompted us to establish a non-human primate disease model. In marmosets, #003-102 Ab ICV infusion (10 µg/h) evoked robust behavior and motor abnormalities, although the CSF concentration of #003-102 Ab in marmosets (10 – 20 µg/mL) was higher than that reported in anti-NMDAR encephalitis patients[14] (hypothetical #003-102 Ab equivalent CSF autoantibody concentrations in patients were 0.10 – 0.43 µg/mL except for one outlier).

In the present study, we used #003-102 Ab as a standard pathogenic anti-NMDAR autoantibody and evaluated the blocking activity of ART5803 against the pathogenic effects of #003-102 Ab. Pathogenic #003-102 Ab (bivalent IgG) induced NMDAR internalization in NMDAR-expressing HEK293 cells and reduced NMDAR surface expression and function ($Ca^{2+}$ influx). Furthermore, we observed #003-102 Ab-induced NMDAR internalization in mouse hippocampal neurons, accompanied by a reduction in spine size. In our marmoset model, ICV infusion of #003-102 Ab induced robust NMDAR expression reduction and behavior/motor abnormalities. Importantly, all neural and behavioral dysfunctions related to hypo-function of NMDAR induced by #003-102 Ab were blocked and restored by concurrent administration of ART5803.

Our study provides compelling evidence that ART5803, as a monovalent IgG antibody, lacks any pathogenic activity on NMDAR in both cellular and neuronal models. Specifically, ART5803 did not induce NMDAR internalization or exhibit agonist or antagonist effects in NMDAR-expressing HEK293 cells. Similarly, in mouse hippocampal neurons, ART5803 administration did not result in NMDAR internalization or spine size reduction. When administered alone via continuous ICV at 10 ug/h or IP injections up to 800 mg/kg twice a week in marmosets for 2 weeks, ART5803 did not induce any observable behavioral abnormalities.

While it is widely accepted that the pathogenicity (NMDAR internalization) of anti-NMDAR encephalitis autoantibodies is driven by their binding to selective epitopes in the GluN1-NTD, it is critical to acknowledge autoantibodies may exist polyclonally and bind to different epitopes in different parts of NMDAR. During the review process of this manuscript, two independent studies were published reporting epitopes of anti-NMDAR autoantibodies[54,55]. Both studies demonstrated that monoclonal antibodies from anti-NMDAR encephalitis patients bind to GluN1-NTD, which corroborates previous findings[15], but also discovered epitopes in both R1 and R2 subdomains. While our data and Michalski et al.[55] demonstrated ART5803 and #003-102 Ab bind to R2 subdomain, these recent studies also discovered autoantibodies binding to R1 subdomain. In the present study, we demonstrated that ART5803 blocked NMDAR internalization in HEK293 cells induced by all patient derived samples tested (monoclonal antibodies, sera and CSF). Given the presence of R1 binding autoantibodies in patients and their NMDAR internalization activities[54,55], we propose a hypothetical mechanism of action in which ART5803 blocks the crosslinking of adjacent NMDARs regardless of epitope positions in the GluN1-NTD; either by direct binding inhibition through epitope competition or steric hindrance, or by binding inhibition of autoantibodies to two adjacent NMDARs by the bulky ART5803 structure attached perpendicularly to R2 subdomain (Supplementary Fig. S11).

We have developed ART5803 based on the long-standing theory that "NMDAR crosslinking" and subsequent "NMDAR internalization" causes pathogenicity leading to anti-NMDAR encephalitis disease manifestations[1,10–12]. Recent publications have proposed that anti-NMDAR autoantibodies may have other effects on NMDARs than the classical crosslinking-internalization activity, such as rapid alternation of extrasynaptic NMDAR trafficking[56] and direct inhibition of NMDAR ionotropic activities[55]. Neural plasticity and physiological adaptation involving NMDARs has been well documented[57,58]. We do not yet know which specific effects of autoantibodies on NMDARs (crosslinking-internalization, extrasynaptic trafficking, direct ionotropic inhibition, or others) are responsible for driving anti-NMDAR encephalitis disease manifestation. Given the practical challenges in assessing these potential autoantibodies' effects in patients, ART5803 could serve as a valuable tool to bridge preclinical studies and clinical investigations. The therapeutic efficacy of ART5803 in patients may address some of these unanswered questions.

Although monoclonal antibody approaches have traditionally lagged behind small molecule (organic chemistry based) approaches in the field of CNS drug discovery, the recent success of anti-amyloid-beta monoclonal antibodies, such as lecanemab and donanemab, for Alzheimer's disease[59] provides both hope and precedent for targeted approaches using monoclonal antibodies to treat CNS disorders. While the precise human doses of ART5803 must be determined based on actual human PK data, our human PK simulations from cynomolgus monkey PK data and precedent clinical data on monoclonal antibodies for Alzheimer's disease[60] and Parkinson's disease[61,62] suggest that brain (CSF) penetration of monoclonal antibodies is attainable using standard dosing regimens, such as IV injection/infusion.

Based on the unique mechanism of action, it is conceivable to co-administer ART5803 with immunosuppressive therapies. ART5803 may constitute a therapy to alleviate acute symptoms effectively and swiftly but may then be subsequently or concomitantly applied with immunosuppressive therapies, such as steroids and rituximab for the maintenance of efficacy. Additionally, ART5803 may be a unique treatment option for relapse and/or treatment-resistant conditions caused by sustained anti-NMDAR autoantibody production. The clinical potential of ART5803 will be addressed in future clinical trials. Although scarcity of marmosets and samples from a rare patient population restricted the sample size and robust statistical analyses in some studies, given the profound and rapid efficacy of ART5803 observed in our marmoset model, we believe that ART5803 could become a promising therapy for patients with anti-NMDAR encephalitis and potentially for other CNS disorders caused by anti-NMDAR autoantibodies.

## Methods

### Test antibodies
Therapeutic antibody ART5803, pathogenic anti-NMDAR auto-antibodies (clones; #003-102 Ab, #007-124 Ab, #007-168 Ab, #007-169 Ab, #008-218 Ab), anti-keyhole limpet hemocyanin (KLH) antibody (isotype control human IgG1), and reference pathogenic anti-NMDAR autoantibody 502 Ab were prepared by Astellas Pharma Inc. (Tokyo, Japan). The synthesis methods and amino acid sequence of ART5803 are published in patent WO2021241616A1. Fab ART5803 was prepared using the Pierce Fab Preparation Kit (Thermo Fisher Scientific) according to the manufacturer's instructions. Bivalent ART5803 was designed using the Fab domain of ART5803 fused to human IgG1 Fc with LALA mutations (L234A/L235A). The bivalent form of ART5803 was manufactured by Twist Biosciences.

Amino acid sequences of the variable region of 502 Ab and pathogenic anti-NMDAR autoantibodies #003-102 Ab, #007-124 Ab, #007-168 Ab, #007-169 Ab, and #008-218 Ab are published in WO2021241616A1 and WO2017029299A1, respectively. The signal sequence and constant region of human IgG were combined with each variable region. The antibodies were expressed in ExpiCHO-S cells (Thermo Fisher Scientific) and then purified by affinity chromatography. Size exclusion chromatography was used for further purification. For Fig. 2, #003-102 Ab was manufactured by LakePharma, and control human IgG1 antibody was purchased from BioLegend.

Given ART5803 is still in the early phase clinical development to understand the safety of this molecule in humans, and importantly the regulatory requirement to monitor and report adverse events, we are obligated to restrict the distribution of ART5803 to protect patient safety and assure data interpretability. The sequence information for ART5803 is publicly available through our patent WO2021241616A1, enabling interested researchers to synthesize the molecule independently if desired. We anticipate that, following the completion of the clinical safety assessments, ART5803 will be made available to interested researchers to facilitate further scientific exploration and collaboration.

### Human samples
Seven anti-NMDAR encephalitis patients' serum and CSF (mRS≥3 at the time of collection) were obtained with written informed consent from the patients or their proxies under collaboration with Kitasato University, Japan. All experiments with human samples were approved by the Institutional Review Board (IRB) of Kitasato University (approval #C18-297) and/or Astellas Research Ethics Committee (approval #A190154). To protect patient identities in this report, patient serum and CSF were given new IDs #1-7, with matching IDs indicative of samples drawn from the same patient. Healthy human pooled serum was purchased from Cosmo Bio Co. LTD (FDA Establishment License #439), and healthy human pooled CSF were purchased from BioIVT LLC (IRB approval #20172667).

### Reagents
Tetracycline, memantine and MK-801 were purchased from Sigma-Aldrich. High sensitivity streptavidin-HRP was purchased from Thermo Fisher Scientific. Phycoerythrin (PE) and allophycocyanin (APC) conjugated goat anti-human IgG were purchased from Jackson ImmunoResearch Inc. Human Fc receptor binding inhibitor, LIVE/DEAD Fixable near IR and violet viability kit, NMDA were purchased from Invitrogen. BM Chemiluminescence ELISA Substrate (POD) was purchased from Roche. FLIPR calcium 6 assay kit loading buffer was purchased from Molecular Devices.

### GluN1 protein preparation for ELISA and Biacore
N-terminal domain (residues 1-393) of human GluN1 (GluN1-NTD) with a C-terminal His-tag was expressed in Expi293F cells. Purification was performed in two steps: Ni-affinity chromatography, followed by size exclusion chromatography.

### GluN1 protein preparation for crystallization
N-terminal domain (residues 1-393) of human GluN1 (GluN1-NTD) with a C-terminal His-tag was expressed in Expi293F cells. GluN1-NTD was purified by Ni-affinity chromatography and digested by endoglycosidase H (Sigma-Aldrich). Size exclusion chromatography was used for further purification. Purified GluN1-NTD was prepared in a buffer of 20 mM Tris-HCl, pH 7.5, and 150 mM sodium chloride. #003-102 Ab Fab' antibody was expressed using the GS Gene Expression System®, using the CHOK1SV® cell line in combination with double gene GS® expression vectors (Lonza). #003-102 Ab Fab' antibody was purified by Protein G affinity chromatography (Kaneka) and subsequently purified by cation exchange chromatography. For crystallization, #003-102 Ab Fab' antibody was prepared in a buffer of 20 mM Tris-HCl, pH 7.5 and 150 mM sodium chloride.

## Cells for NMDAR internalization and functional (Ca²⁺ influx) assays

NMDAR expressing HEK293 cells (HEK293 cells expressing tetracycline-inducible human NMDAR subunit GluN1/GluN2B) were purchased from Charles River. Non-NMDAR expressing cell line, T-REx293, was purchased from Invitrogen. Cells were passaged according to the manufacturer specifications. To induce high NMDAR expression for internalization assays, cells were cultured in Induction Medium (IM) consisting of Neurobasal Medium (Life Technologies Corporation) with 10% dialyzed heat-inactivated fetal bovine serum (FBS), 50 U/mL penicillin-streptomycin, 2 μg/mL tetracycline and 0.2 mM memantine at 37 °C, 5% $CO_2$. For functional assays, cells were cultured in IM without tetracycline.

## Binding activity assays to GluN1 protein by Biacore

Affinity and binding kinetics were evaluated using the Biacore T200 (GE healthcare). ART5803 and #003-102 Ab were adjusted to 10 μg/mL in HBS-EP buffer and immobilized on a Series S Sensor Chip CM5 using an Antibody Capture kit (GE healthcare) at a flow rate of 10 μL/min for 50 sec, according to manufacturer's instructions. Human GluN1-His was adjusted to 0.50 μg/mL with HBS-EP + Buffer and serially diluted 2-fold across 8 concentrations (0.50 to 0.0039 μg/mL). Dissociation was performed with HBS-EP + buffer at 50 μL/min for 5 min, followed by regeneration with 3.0 M $MgCl_2$ at 20 μL/min for 30 sec. $K_D$ was calculated with a 1:1 binding model by BIAevaluation software (GE healthcare).

## X-ray crystallography

GluN1-NTD and #003-102 Ab Fab' were mixed with a molar ratio of 1:1 and incubated for 1 h at room temperature. Crystals were obtained by sitting drop vapor diffusion method using a reservoir solution of 0.1 M MES or HEPES, pH 6.5-7.5, and 6-8 %w/v PEG20000. X-ray data were collected at SPring-8 BL32XU beamline (Hyogo, Japan). X-ray diffraction data from four crystals were processed and merged using the KAMO system with XDS. Initial phases were determined by molecular replacement using Phaser in the CCP4 suite with the coordinates of rat GluN1-NTD (PDBID: 3Q41) as a search model. The model was built manually with Coot and refinement was conducted using REFMAC5 in the CCP4 suite. The final structure of the complex between GluN1-NTD and #003-102 Ab Fab' antibody was deposited in the protein data bank (PDBID: 8ZH7). Statistics are summarized in Supplementary Table S10. Supplementary Fig. S12 shows an image of a portion of the electron density map of GluN1-NTD and #003-102 Ab Fab' antibody (2Fo−Fc ma, contour level: 1.0σ). Image generation and structural analysis completed in The PyMol Molecular Graphics System, Version 3.0 Schroedinger, LLC.

## Hydrogen Deuterium Exchange - Mass Spectrometry (HDX-MS)

HDX-MS of #003-102 Ab:GluN1-NTD and ART5803:GluN1-NTD complexes were performed at U-Medico (Osaka, Japan). The antigen was diluted with D-PBS(-). The antigen-antibody was mixed and diluted with D-PBS(-). After D-PBS(-) was dried under reduced pressure, an equal volume of deuterated water ($D_2O$) was added. These samples were diluted 20-fold by $D_2O$. Then these mixtures were diluted 2-fold by Quench buffer (100 mM phosphoric acid, 150 mM TCEP, 4 M guanidine HCl, pH 2.1). Deuterium exchange times were 0 sec, 30 sec, 1 min, 10 min, 1 h and 4 h, respectively. After mixing, samples were immediately injected into the Waters HDX system. The samples were digested by an Enzymate Pepsin Column (Waters). The flow rate during sample digestion was 50 μL/min for 6 min. The digested peptides were desalted on VanGuard pre-column (Waters). Peptides were separated by changing the concentration of acetonitrile containing 0.1% formic acid from 8% to 40% in 9 min. Peptides were measured using a Synapt G1 mass spectrometer (Waters). Data were acquired using MSE mode in the m/z range of 100-2000. Obtained data were analyzed using

Waters ProteinLynx Global SERVER (PLGS) to detect signals of peptides. All measurements were repeated in triplicate. Peptides commonly detected in antigen and antigen-antibody complexes were analyzed using DynamX software (Waters). Statistical analysis was performed as described in Hageman et al., 2019[63]. Image generation and structural analysis completed in The PyMol Molecular Graphics System, Version 3.0 Schroedinger, LLC.

## Assessment of ART5803 agonistic and antagonistic activity on NMDAR by FLIPR

NMDAR-expressing HEK293 cells in IM ($1 \times 10^6$ cells/well) were plated on a poly-D-lysine 96-well flat-bottom plate (Corning Inc.). After 16-24 h of incubation at 37 °C, 5% $CO_2$, FLIPR Calcium 6 Assay Kit Loading Buffer was prepared according to the manufacturer's instructions. Briefly, FLIPR loading buffer was added to each well, and further incubated for 2 h. ART5803 or the human IgG isotype control antibody (0.010, 0.10, 1.0, 10 μg/mL as final concentrations), MK-801 (10 μM as final concentration), or assay buffer only were added by FLIPR 3 (Molecular Devices) and fluorescence data was acquired over 5 min. Sequentially NMDA (30 μM as final concentration in assay buffer) or assay buffer only was added by FLIPR 3 and fluorescence data was acquired over 5 min. Max-min fluorescence values were calculated by ScreenWorks (Molecular Devices). Data from each independent replicate was reported as $RFU_{(max-min)}$ mean.

## Competitive activity assay with biotinylated #003-102 Ab by ELISA

MaxiSorp 384-well ELISA plates (Thermo Fisher Scientific) were coated with 1.0 μg/mL of human GluN1 NTD protein at 4 °C overnight. After washing with TBS-T (Nippon Gene Co Ltd.), Blocking One (Nacalai Tesque Inc.) was added to block wells at room temperature for 1 h. Antibodies were serially diluted (0.10 to 100,000 ng/mL) and incubated for 20 min. Biotinylated #003-102 Ab (generated by Biotin Labeling Kit NH2 according to manufacturer's instructions (Dojindo)) was diluted to 600 ng/mL in 5% Blocking One/TBS-T and incubated at room temperature for 1 h. Streptavidin-HRP was diluted to 1:8000 in 5% Blocking One/TBS-T and incubated for 30 min. BM Chemiluminescence ELISA substrate POD was prepared and used according to manufacturer's instructions. The chemiluminescence was measured with EnVision (PerkinElmer). One hundred percent inhibition was determined by competition of 600 ng/mL biotinylated #003-102 Ab with 100,000 ng/mL unconjugated #003-102 Ab and 0% inhibition was determined by binding of 600 ng/mL biotinylated #003-102 Ab in the absence of competitor and control antibodies.

## Assessment of the effect of ART5803 and pathogenic autoantibody #003-102 Ab on NMDAR internalization

Tetracycline inducible NMDAR-expressing HEK293 cells were used for this study and expression was induced with IM for 16–24 h. All incubations were performed at 37 °C, 5% $CO_2$, unless otherwise stated. Cells were harvested and seeded at $1.5 \times 10^5$ cells/well in tetracycline free IM and incubated for 24 h. All antibody dilutions were performed in tetracycline free IM. Serial dilutions of ART5803, Fab ART5803, Bivalent ART5803, IgG1 isotype control antibody, or #003-102 Ab, were added to cells for a final concentration between 0.0003-10 μg/mL and incubated for 2 h. Cells without antibody treatments in IM were also prepared for the assessment of 100% and 0% NMDAR surface expression. After incubation with antibodies, cells were harvested and incubated with human Fc receptor binding inhibitor (1:50; Invitrogen, 14-9161-73) and LIVE/DEAD Fixable violet viability kit (1:667, Invitrogen, L34964) for 15 min. After washes, cells were fixed with 4% PFA for 20 min. Cell surface NMDARs treated with ART5803, bivalent ART5803, #003-102 Ab and isotype control were stained with 5.0 μg/mL of ART5803 or human IgG1 isotype control antibody, respectively. For Fab ART5803 incubated cells, NMDAR cell surface stain was done with 5.0 μg/mL of

Fab ART5803 or human IgG1 isotype control antibody, respectively. All antibody staining were performed for 30 min and incubated with APC conjugated goat anti-human HC/LC IgG (1:100) for 30 min. Then, the APC signal detection was performed by using Novocyte 2060, Novo Express v 1.5.0 (Agilent Technologies). Flow cytometry data was analyzed by FlowJo (BD Biosciences). Expression levels of NMDARs on the cell surface were calculated after normalization to the mean fluorescence intensity (MFI) of ART5803 staining as 100% surface expression and the MFI of human IgG isotype control antibody Ab staining as 0% surface expression. Average GluN1 surface expression from 4–5 independent experiments were graphed using GraphPad Prism software. Supplementary Fig. S13 shows the flow cytometry gating strategy.

### Assessment of ART5803 block and restoration of NMDAR Internalization Assay using pathogenic autoantibody #003-102 Ab

NMDAR-expressing HEK293 cells were harvested and seeded at $1.5 \times 10^5$ cells/well in IM. Cells were incubated at 37 °C, 5% $CO_2$ for a total of 6 h. All antibody dilutions were also performed in IM. To test whether ART5803 could block receptor internalization, #003-102 Ab (1.0 μg/mL as final concentration) and ART5803 (0.0003-10 μg/mL as final concentration) were co-incubated with cells for 2 h. To test whether ART5803 could rescue receptor internalization, #003-102Ab (1.0 μg/mL as final concentration) was incubated for 2 h, then ART5803 (0.0003-10 μg/mL as final concentration) was co-incubated for 2 h. Cells without antibody treatments were also prepared for the assessment of 100% and 0% NMDAR surface expression. The following steps were performed on ice or at 4°C. After incubation with antibodies, cells were harvested and incubated with human Fc receptor binding inhibitor (1:50; Invitrogen, 14-9161-73) and LIVE/DEAD Fixable violet viability kit (1:667; Invitrogen, L34964) for 15 min. Cells were fixed with 4% PFA for 20 min. Cell surface NMDARs were stained with 5.0 μg/mL of ART5803 or human IgG isotype control antibody for 30 min and incubated with APC-conjugated goat anti-human Fc IgG (1:100) at 4°C for 30 min. Then, the APC signal detection was performed by using Novocyte 2060, Novo Express v 1.5.0 (Agilent Technologies). Flow cytometry data was analyzed by FlowJo (BD Biosciences). The MFI of T-REx-293 cells was subtracted from corresponding treated NMDAR expressing HEK293. Expression levels of cell surface NMDARs were calculated after normalization to the MFI of ART5803 staining as 100% surface expression and the MFI of human IgG isotype control antibody Ab staining as 0% surface expression. Average GluN1 surface expression from 3-4 independent experiments were graphed using GraphPad Prism software.

### Assessment of ART5803 effect on NMDAR hypofunction evoked by #003-102 Ab by FLIPR

The pathogenic autoantibody #003-102 Ab or the human IgG1 isotype control antibody (1.0 μg/mL as final concentration for each antibody) were prepared in IM. NMDAR-expressing HEK293 cells in IM ($1.5 \times 10^5$ cells/well) were added with #003-102 Ab or control antibody to a poly-D-lysine 96-well flat-bottom plate (Corning Inc.). The plate was incubated at 37 °C, 5% $CO_2$ for 15 min. ART5803 or the control antibody in IM (0.01, 0.1, 1.0, 10 μg/mL as final concentrations) were added to the wells and further incubated for 6 h. After incubation and aspiration, FLIPR Calcium 6 assay kit (Molecular Devices) was performed according to manufacturer's instructions. Briefly, FLIPR loading buffer was added to wells and further incubated for 2 h. Stimulations were performed with NMDA (30 μM as final concentration). Fluorescence intensity, indicating changes in intracellular calcium levels, was detected using FLIPR 3 (Molecular Devices). Max-min RFU were calculated by ScreenWorks (Molecular Devices). Data from each independent replicate was reported as $RFU_{(max-min)}$.

### Mouse hippocampal organotypic slice culture assay

**Animals.** Male and female wild-type mice (C57BL/6 J) were purchased from Jackson Laboratory, Envigo, or bred in-house (University of California Davis, Davis, CA, USA). All experimental protocols were approved by the University of California Davis Animal Care and Use Committee (#23556). Mice were housed in conventional rodent cages, under an environmental temperature range of 20-26°C and 30-70% humidity.

**Preparation and transfection of organotypic slice cultures.** Organotypic hippocampal slice cultures were prepared from decapitated postnatal day 7-8 C57BL/6 J wild-type mice of both sexes, as described[64]. Slice cultures were transfected by biolistic particle-mediated gene delivery (gene gun) at DIV 7-12, as described[65]. Bullets contained 20 μg of SEP-GluN2A and 10 μg of GluN1, and 5.0 μg of TdTomato coated onto 8.0 mg of 1.6 μm gold beads[27]. Slice cultures were incubated at 35 °C for 7 days to allow for stable steady-state expression.

**Time-lapse two-photon imaging.** Cultures were maintained at 35 °C with 5% $CO_2$ except during brief imaging at room temperature. Slices were imaged in culture medium: MEM Eagle medium 8.4 g/L, horse serum 20%, $CaCl_2$ 1 mM, $MgSO_4$ 2.0 mM, L-Glutamine 1.0 mM, ascorbic acid 0.00125%, insulin 1.0 mg/L, D-glucose 13 mM, sodium bicarbonate 5.2 mM, HEPES 30 mM in $ddH_2O$; pH 7.28, 320 mOsm. Image stacks (512 × 512, 1.0 μm z-steps) of two secondary or tertiary CA1 pyramidal neuron dendritic segments were collected on a custom two-photon microscope[66] with a pulsed Ti:Sapphire laser (930 nm, 2.0 mW at the sample; Spectra Physics). Data acquisition was controlled by ScanImage[67]. Images are shown as maximum projections of 3D image stacks after a median filter (3 × 3). Only neurons that survived all three time points were analyzed.

**Antibody treatment.** Antibodies (each at 25 μg/mL, final concentration) were applied to slice cultures immediately after baseline images for all except rescue experiments. For rescue experiments, #003-102 Ab were added after baseline images and ART5803 was administered after the 3 h images.

**Image analysis.** Images were analyzed blind to treatment conditions, as described[66]. Briefly, spines were identified from red fluorescence (TdTomato cell fill) blind to green fluorescence (SEP-GluN2A) levels. Boxes around spines and spine necks were drawn in the red channel and transferred to the green channel. If unspecific fluorescence (e.g. debris clearly not belonging to the spine) was present in any of the boxes, the position was adjusted, or a spine was excluded from the analysis. Integrated red (spine size) and green (surface NMDAR) fluorescence were measured from the target spine head and nearby background was subtracted by drawing a box next to the target spine. Data are shown as mean ± SEM. GraphPad Prism software was used to calculate ordinary paired one-way ANOVA with Tukey's multiple comparisons.

### Anti-NMDAR encephalitis disease model in marmoset

**Animals.** For ART5803 efficacy through ICV infusion evaluation, adult common marmosets (12 males, ≥1 year old) were purchased from Shin Nippon Biomedical Laboratories, LTD (Kagoshima, Japan). Animals were singly housed in a cage (44 x 44 x 61 cm) with stainless steel grid doors, a horizontal perch, and mezzanine floor during behavioral recordings. This animal experimental procedure was approved (D-T19088) by the Institutional Animal Care and Use Committee of Astellas Pharma Inc. (Tokyo, Japan). The Tsukuba Research Center of Astellas Pharma Inc. is accredited by the Association for Assessment and Accreditation of Laboratory Animal Care International. For ART5803 IP injection evaluation, adult common marmosets (15 males

and females, ≥1 year old) were performed at Shin Nippon Biomedical Laboratories, LTD (SNBL; Kagoshima, Japan). Animals were singly housed in a cage (49 x 46 x 49 cm) with stainless steel grid doors, a horizontal perch and mezzanine floor during behavioral recordings. This animal experimental procedure was approved (IACUC596-006) by the Institutional Animal Care and Use Committee of SNBL (Kagoshima, Japan). The SNBL is accredited by the Association for Assessment and Accreditation of Laboratory Animal Care International.

**ICV infusion of #003-102 Ab.** For both ART5803 ICV and IP efficacy studies, #003-102 Ab was administered through continuous ICV infusion as follows: Marmosets were equipped with a respirator (SHINANO manufacturing CO., LTD, SN-480-7) and anesthesia was induced by isoflurane (Mylan Pharmaceutical, 871119). A stereotaxic frame was used to immobilize the head (NARISHIGE Group, SR-6C-HT) and the cannula (Plastics One Inc, Roanoke, 813220PSPCLC) was placed [A / P: 3.5 mm, M / L: ± 0.0 mm, D / V: −6.95 mm (from dura mater)] with reference to the Marmoset brain map (Stereotactic Atlas of the Marmoset Brain, NCBI Bookshelf ID: NBK55612). Next, the cannula was connected to a micro-infusion pump (iPRECIO, SMP-200), and the pump was set in a jacket pocket (manufactured in-house) that minimizes interference of behavioral observation. For one week after cannula placement in the ventricle, PBS was ICV infused at a flow rate of 2.0 μL/h using a micro-infusion pump. Then, it was replaced with a micro-infusion pump filled with pathogenic #003-102 Ab (5,000 μg/mL), at a 2.0 μL/h (10 μg/h) flow rate.

**ICV administration of ART5803.** After 14 days of continuous #003-102 Ab infusion, the micro-infusion pump was replaced by a mixture of pathogenic antibody and therapeutic antibody (ART5803, $n = 9$) or control Ab ($n = 3$) in equal amounts for each (2500 μg/mL) and delivered at a flow rate of 4.0 μL/h (10 μg/h) for 14 days.

**IP administration of ART5803.** #003-102 Ab was continuously infused for a total of 21 days. On day 6 of #003-102 Ab ICV infusion, marmosets showing robust abnormal behaviors were divided into two treatment groups. One treatment group ($n = 8$) received ART5803 400 mg/kg via IP injections twice a week for 2 weeks starting on Day 7. The other treatment group ($n = 7$) received vehicle control (20 mM sodium acetate, 140 mM arginine, 0.02% w/v polysorbate 80, pH 5.0) via IP injection twice a week for 2 weeks starting on Day 7.

**Abnormal behavior and motor scoring.** To quantify the degree of behavioral and motor deficits, a modified ARS was derived from the modified Unified Parkinson's Disease Rating Scale (UPDRS) previously established in marmosets[29] which movement and behavioral dysfunctions resemble anti-NMDAR encephalitis patients. This 22-point scale evaluates behavior (attention, motivation, fear/anxiety), and motor movements (speed of movement, motor coordination, jumping) (Supplementary Table S4). On behavior examination days, animals were individually filmed in the cage for at least 5 min and all ARS evaluations were conducted by a blinded marmoset behavior observation expert reviewing the video recordings. During the recording session for each marmoset, the caregiver dangled an enrichment toy (cable tie) through the cage in front of the animal, then after removing the enrichment toy, the caregiver similarly presented a reward (small snack).

For the ART5803 ICV efficacy study, examination days were before ICV infusion of pathogenic antibody (day 0), 14 days after ICV infusion of pathogenic antibody (day 14) and 14 days after mixture of pathogenic antibody with therapeutic antibody or control antibody (day 28). For the ART5803 IP efficacy study, examinations day were on day 0 (prior to #003-102 Ab ICV infusion), day 6 of #003-102 Ab ICV infusion (prior to ART5803 IP or vehicle control injection), day 14

(1 week post ART5803 or vehicle treatment) and day 21 (2 weeks post ART5803 or vehicle treatment).

**Immunohistochemistry.** For all animals that were ICV infused with #003-102 Ab and treated IP with ART5803 or Vehicle as previously described, and additionally 3 naive animals were anesthetized by inhalation of isoflurane, received an IV bolus of heparin sodium (200 IU/kg) into the tail vein, and perfused via the left cardiac ventricle with ice cold 100 mL PBS to expel the blood. The brain was divided into left and right hemispheres. The cerebral cortex (whole) from the right hemisphere was dissected and immediately frozen in liquid nitrogen and stored in a deep freezer for western blot analysis. The left hemisphere was fixed in 4% PFA in 0.1 mol/L phosphate buffer on wet ice for 2 h, then desiccated in 25% sucrose buffer at 4°C overnight. This hemisphere was further divided into two or three sagittal sections and embedded in Optimal Cutting Temperature Compound (Sakura Finetek Japan Co., Ltd, 4583). Samples were stored in Histo-tek Hyfluid cooling refrigerant (Sakura Finetek Japan Co., Ltd., PINO-HF) at −80°C until processed.

For human IgG, GluN1 (NMDAR) and GluR2 (AMPA receptor) staining, brains were cryosectioned at 6 μm onto slides and fixed in cold acetone for 10 min. Slides were washed with cold PBS. Subsequent wash steps were performed similarly unless otherwise stated. Endogenous peroxidase activity was quenched with Peroxidase-blocking solution (1:100 ratio of hydrogen peroxide to methanol) at room temperature for 30 min. Slides were then washed and blocked with Protein Block Serum Free Solution (Agilent Technologies, Inc., X0909) for 30 min. Primary antibodies for anti-GluN1 (Merck KGaA, MAB363) and anti-GluR2 (Merck KGaA, MAB397) were in diluted Antibody Dilutant (Agilent Technologies, Inc., S3022) 500 and 5000-fold, respectively. Primary antibody goat anti-human IgG (Southern Biotechnology Associates, Inc., 2049-05) was diluted 50-fold in 1x PBS. Antibodies were incubated on slides at room temperature for 1 h. Slides stained for anti-GluN1 and anti-GluR2 were then treated with Histofine simple stain MAX-PO(M) (Nichirei Bioscience Inc., 424131) at room temperature for 30 min. Slides were washed then all slides were treated with DAB substrate solution (Agilent Technologies, K3468) at room temperature for the following lengths of time: anti-human IgG, 1.5 min; anti-GluN1, 1 min; anti-GluR2 30 sec. Reaction was stopped by immersion in cold PBS. Slides were then washed under running tap water for 3 min and immersed in distilled water for 1 min, and subsequent washes were performed similarly. Tissues were counterstained at room temperature with 1x Mayer's Hemalum solution (Marck KGaA, 1.09249.0500) and washed. Lastly, tissues were dehydrated with 70% alcohol for 3 min and 100% alcohol for 3 min, followed by clearing in xylene for 3 min, then cover slipped. Slides were scanned with NanoZoomer 360 and NDP view2 software version 2.9.29 at 40x magnification (Hamamatsuo Photonics K.K.).

GluN1 and GluR2-stained whole slide images were analyzed using QuPath version 5.1, focusing on the segmentation and quantification of superpixels. Hematoxylin and DAB staining patterns were separated by the QuPath algorithm. The prefrontal cortex was chosen as the region of interest, as across all tissues as it was the least folded or torn and contained the least staining irregularities. A square annotation area was used, which was approximately 450,000 μm², Superpixel segmentation was performed according to the methods in Morris et al.[35].

The DAB mean threshold for assigning superpixels as positive or negative was determined using histograms generated from three representative regions of interest (ROIs) within the prefrontal cortex. Each DAB mean histogram generated by QuPath for each ROI was segmented by identifying the value at the peak of the histogram, representing the most frequent DAB mean intensity, and the right tail end value, representing the highest DAB intensity observed. The range between these two points were divided into three equal bins, corresponding to high, medium, and low intensity thresholds within each

ROI. This approach allowed for a systematic and representative threshold determination based on the distribution of staining intensities. Threshold values for each intensity level were averaged across 3 ROIs to obtain a more representative threshold for analysis. These threshold values were subsequently applied to all tissues that were stained in parallel. For each tissue analysis, 3 ROIs were drawn, and the analysis was applied. The percentage positive superpixels for each tissue were averaged and reported. Heatmaps were also generated in QuPath based on DAB mean intensity only to represent staining intensity, not necessarily positivity.

**Western blotting.** Proteins were extracted from frozen cerebral cortices (whole) using total protein extraction kit (Merck, 2140) according to the manufacturer's instructions. The protein concentration of samples was determined using the Pierce Dilution-Free Rapid Gold BCA Protein Assay (Thermo Fisher Scientific, A55860). Samples were mixed with 2× Sample Buffer Solution with 2-ME for SDS-PAGE (Nacalai Tesque Inc, 30566-22) and denatured at 95 °C for 5 min. Denatured samples (5 μg per lane) were subjected to SDS-PAGE and transferred to an Immobilon-P membrane (Merck, IPVH00010). After blocking with Blocking One (Nacalai Tesque Inc.), the membranes were probed with the following primary antibodies at 4 °C overnight: mouse anti-GluN1 (1:1000; Merck, MAB363), rabbit anti-GluR2 (0.5 μg/mL; Merck, AB1768-I), rabbit anti-PSD95 (1:1000; Abcam, ab18258), mouse anti-GAPDH (1:1000, Fujifilm, 016-25523). All antibodies were diluted in Can Get Signal Immunoreaction Enhancer Solution (TOYOBO). After washing with Tris-buffered saline containing 0.05% Tween 20 (TBS-T), the membranes were incubated with HRP-linked anti-mouse IgG or anti-rabbit IgG antibodies (1:5000; Cell Signaling Technology) at room temperature for 1 h. After washing with TBS-T, chemiluminescent signals were detected, and bright-field images were acquired using the Chemi-Lumi One L (Nacalai Tesque Inc.) and Amersham Image-Quant800 (Cytiva). The membranes were stripped with WB Stripping Solution (Nacalai Tesque Inc.) at room temperature for 15 min, blocked again, and re-probed. The membranes were then prepared and developed as described above. For normalization, the membranes were stained with Ponceau-S Staining Solution (Beacle) according to the methods in Sander et al.[68] Protein bands from western blot analyses were quantified by densitometry using ImageJ software for graphical representation. Each protein band was normalized against the total protein with ponceau stain. For some figures, unrelated lanes were cropped out. Full size images are provided in Supplementary Fig. S10.

**NMDAR internalization assay using a monoclonal autoantibody mixture (rescue of NMDAR surface expression by ART5803)**
The final concentrations of pathogenic anti-NMDAR monoclonal antibodies in mixture were adapted from Ly et al. 2018 (patient 2-like mix) and made as follows: #003-102 Ab 0.43 μg/mL, #007-168 Ab 4.63 μg/mL, and #008-218 Ab 0.95 μg/mL (final concentrations). Antibodies were prepared in IM. Tetracycline induced NMDAR expressing HEK293 in IM ($1.5 \times 10^5$ cells/well) were added with patient 2-like mix to a 96 well plate. In parallel, IM was added to untreated cells designated for control staining (100% and 0% expression) The plate was incubated at 37 °C, 5% CO$_2$ for 18 h. ART5803 (0.00017–10 μg/mL as final concentration) in IM and further incubated for 30 h. Cells were harvested on ice and NMDARs on the cell surface were stained with ART5803 (5.0 μg/mL) or control human IgG antibody (5.0 μg/mL), and PE conjugated goat anti-human IgG (1:100). The PE signal detection was performed by using FACS Verse (BD Biosciences). FlowJo (BD Biosciences) was used for the analysis of flow cytometry data. The MFI of untreated cells with NMDAR surface stained with ART5803 was regarded as 100% surface expression. The MFI of untreated cells stained with control antibody (background) was regarded as 0% surface expression.

**NMDAR internalization assay using monoclonal autoantibody mixtures (Blockade of NMDAR internalization by ART5803)**
Cell incubation conditions were 37 °C, 5% CO$_2$. Tetracycline induced NMDAR expressing HEK293 cells were harvested and plated in a 96-well plate. ART5803 (0.001–10 μg/mL as final concentration) serially diluted in IM without tetracycline (IM tet-) was added to the cells and preincubated for 2 h. Each concentration of serially diluted ART5803-treated cells were prepared in duplicate cell wells. In parallel, IM tet-only was added to untreated cells designated for control staining (100% expression and 0%).

Monoclonal antibody mixtures (patient 2-6 like mix) were prepared in IM tet- as follows (final concentrations). Patient 2 like mix was prepared by pooling 0.43 μg/mL #003-102 Ab, 0.95 μg/mL #008-218 Ab, and 4.63 μg/mL #007-168 Ab. Patient 3 like mix was prepared by pooling 0.39 μg/mL #003-102 Ab, 0.84 μg/mL #008-218 Ab, 4.00 μg/mL #007-168 Ab, and 100 μg/mL #007-124 Ab. Patient 4 like mix was prepared by pooling 0.34 μg/mL #003-102 Ab, 0.68 μg/mL #008-218 Ab, 3.19 μg/mL #007-168 Ab, and 60.5 μg/mL #007-124 Ab. Patient 5 like mix was prepared by pooling 0.14 μg/mL #003-102 Ab, 0.22 μg/mL #008-218 Ab, 0.91 μg/mL #007-168 Ab, 17.0 μg/mL #007-124 Ab, and 100 μg/mL #007-169 Ab. Patient 6 like mix was prepared by pooling 0.10 μg/mL #003-102 Ab, 0.16 μg/mL #008-218 Ab, 0.63 μg/mL #007-168 Ab, 13.5 μg/mL #007-124 Ab, and 79.9 μg/mL #007-169 Ab. Each prepared mix and human IgG1 control antibody were added to respective cell wells and incubated for an additional 24 h. Cells were harvested into a 96-well round-bottom plate and were added human Fc receptor binding inhibitor (Invitrogen, 14-9161-73) for 15 min. After Fc receptor blocking, cells were stained with 5.0 μg/mL of either ART5803 or anti-KLH IgG1 control antibody for 30 min. After a wash, cells were stained with Phycoerythrin (PE) conjugated goat anti-human IgG (1:100) for 30 min. The PE signal detection was performed by using FACS Verse (BD Biosciences). FlowJo (BD Biosciences) was used for the analysis of flow cytometry data. The percentage of GluN1 surface expression was calculated as described in previous internalization assay sections.

**Detection of autoantibodies in anti-NMDAR encephalitis patients' serum and CSF**
NMDAR-expressing HEK293 and non-expressing cells were incubated with human Fc receptor binding inhibitor (1:50) and LIVE/DEAD Fixable Near-IR dead cell stain (1:6667) for 15 min. After a wash, cells were plated at $1.5 \times 10^5$ cells/well in 96 well plates. To generate the 502 Ab concentration versus fluorescence standard curve, 502 Ab was diluted 10-fold from 10-0.0001 μg/mL as final concentrations in 40% healthy pooled human CSF or 1.0% and 10% of healthy pooled human serum (final percentages) and added to the NMDAR-expressing HEK293 cells. The 502 Ab standard curve was run in parallel with the below experimental conditions.

Patient serum was diluted to 1.0%, and patient CSF was diluted to 40% (final percentages) in 502 Ab standards, and diluted samples were incubated with NMDAR-expressing HEK293 cells for 30 min. After a wash, PE conjugated anti-human IgG (1:100) was incubated for 30 min. After repeated washes, cells were fixed with 4.0% PFA (Fujifilm Wako Chemicals, 163-20145) for 20 min and measured on a FACS Verse flow cytometer (BD Biosciences). FlowJo (BD Biosciences) was used for the analysis of flow cytometry data. Data was reported as the mean of two internal replicates. For both 502 Ab standard curve and patient samples, the MFI from non-expressing TREx293 cells was subtracted from the NMDAR-expressing HEK293. The 502 Ab standard curve was used to derive a pseudo-concentration of patient sample anti-NMDAR antibodies. The CSF/serum ratio was determined for samples by dividing anti-NMDAR antibody CSF concentrations by serum concentrations. Data are reported as the mean of two internal replicates and were graphed in GraphPad Prism software.

### NMDAR Internalization Assay using anti-NMDAR encephalitis patients' serum and CSF

Tetracycline-induced NMDAR-expressing HEK293 cells were harvested and plated at $1.5 \times 10^5$ cells/well in a 96-well plate. To generate the 502 Ab concentration versus percentage NMDAR expression standard curve, 502 Ab was diluted 2-fold from 1.0 to 0.0078 μg/mL as final concentrations in 10% or 40% healthy pooled human CSF or 3% and 10% of healthy pooled human serum (final percentages) and added to the cultured cells. The 502 Ab standard curve was run in parallel with the below experimental conditions.

For the preparation of the patient sample, co-incubation with ART5803, patient CSF was diluted to 40% or 10% and patient serum was diluted to 3% or 10% (final percentages). ART5803 was diluted to final concentrations between 0.0625 and 2.0 μg/mL. Patient samples and ART5803 were co-incubated with the plated cells at a total volume of 50 μL/well at 37°C, 5% $CO_2$ for 20 h. After incubation, cells were dissociated and transferred to the FACS reading plate (Corning). After washing cells, cells were incubated with human Fc receptor binding inhibitor (20 μg/mL; Invitrogen, 14-9161-73) and dead cell stain kit (3:10,000; Invitrogen, L10119) for 15 min. Then, cell surface NMDARs were stained with 5.0 μg/mL of 502 Ab or control Ab and PE-conjugated goat anti-human IgG (1:100). After a wash with flow cytometry buffer, cells were fixed with 4% PFA for 20 min then analyzed. The PE signal detection was performed by using FACS Verse (BD Biosciences). FlowJo (BD Biosciences) was used for the analysis of flow cytometry data. Data are reported as the result of a single test, except for patient #1 CSF which is the mean of 2 independent experiments. The percentage of GluN1 surface expression was calculated as described in previous internalization assay sections. The surface expression levels of NMDAR from the standard curve generated by the known 502 Ab concentrations were used to calculate patient sample (without ART5803 treatment) relative concentrations by assigning 502 Ab concentrations that induced the same level of NMDAR surface expression.

### Cynomolgus monkey single IV PK study

A single-dose IV infusion PK assessment in cynomolgus monkeys was conducted at Labcorp, a facility fully accredited by the Association for Assessment and Accreditation of Laboratory Animal Care. All procedures in the protocol were in compliance with applicable animal welfare acts and were approved by the local Institutional Animal Care and Use Committee (#8515321). The exposure of ART5803 in serum and cerebrospinal fluid (CSF) following a 30 min IV infusion of ART5803 (100 mg/kg and 1000 mg/kg) to male cynomolgus monkeys (3 males for each dose) during a single-dose pharmacokinetic study with a 12 week recovery period was assessed.

### Cynomolgus monkey population PK (PopPK) modeling with human scaling and simulation

ART5803 cynomolgus monkey serum and CSF concentration data were fitted to a two-compartment model with first order clearance. CSF concentrations were fitted to a separate CSF tissue compartment with bi-directional flow and the same clearance as the central compartment. Overall, the final model was a three-compartment model with first-order clearance from both the central and CSF compartments with 6 structural parameters (V, Cl, V2, Cl2, Vcsf, and Qcsf). The final model fit with cynomolgus monkey data was then allometrically scaled to approximate human parameters using the following equations:

$Vh = tvV \times (80/4.42)^1$, $V2h = tvV2 \times (80/4.42)^1$, $Vcsfh = tvVcsf \times (80/4.42)^1$, $Clh = tvCl*(80/4.42)^{0.8}$, $Cl2h = tvCl2 \times (80/4.42)^{0.8}$, and $Qcsfh = tvQcsf \times (80/4.42)^{0.8}$. Eighty kg was used as mean human body weight and 4.42 kg was the average body weight of the 6 animals in the study. Random effects (between subject variability) were incorporated in the

animal model, as the model was a PopPK model, but was not scaled to human. The final human model parameters are as follows. Central Volume (V) = 4570 mL, Peripheral Volume (V2) = 4340 mL, CSF Compartment Volume (Vcsf) = 1910 mL, Clearance (Cl) = 29.8 mL/h, Central to Peripheral Clearance (Cl2) = 12.3 mL/h, and Central to CSF Flow (Qcsf) = 0.0639 mL/h.

### Statistical analysis

The specific statistical tests used are indicated in the figure legends. Statistical and $EC_{50}$ analyses were performed using GraphPad Prism software. The $EC_{50}$ was calculated using sigmoid-Emax nonlinear regression analysis. Pearson's correlation and simple linear regression were used to measure the linearity between patients' CSF and serum concentrations. For comparisons between two groups, statistical significance was assayed using unpaired, two-tailed Student's t-tests. For ex-vivo timepoint comparisons, paired ordinary one-way ANOVA with Tukey's multiple comparisons was used. For in vivo comparisons between pre-post grouped tests, two-tailed, Wilcoxon matched pairs signed rank test was used to analyze changes in ARS scores. Data are represented as mean ± SEM or mean ± SD. For all statistical analyses, $p$ values are indicated in each figure panel.

### Reporting summary

Further information on research design is available in the Nature Portfolio Reporting Summary linked to this article.

## Data availability

The crystal structure data in this study have been deposited in the Protein Data Bank under accession code 8ZH7: https://doi.org/10.2210/pdb8zh7/pdb. Source data are provided with this paper.

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

## Acknowledgements

This work was supported by Arialys Therapeutics Inc. and Astellas Pharma, Inc. We thank members of Shin Nippon Biomedical Laboratories, Ltd., for marmoset studies and Laboratory Corporation of America Holdings (Labcorp) for cynomolgus monkey PK studies.

## Author contributions

A.K., T.K., M.Maeda, S.Y., Y.A., T.Shimomura, M.Anisimova, N.K., K.S., A.R., T.M., K.K., T.Shimada, K.N., N.N., Y.K., A.O., A.S., D.P., I.S., S.P., V.V., S.B., R.K., V.E., and S.J.S. performed experimental studies and data analyses. M.Mitchell, M.Maurer, M.J., J.L., D.Y., H.S., M.Adachi, D.J.H., S.K., K.Z., T.I., P.F., and M.Matsumoto supervised studies, provided critical input, and supported data interpretation. A.K., T.K., K.Z., and M.Matsumoto wrote the paper with input from all co-authors. All authors read and approved the paper.

## Competing interests

A.K., M.Maeda, S.K., and S.B. are consultants, and S.Y., A.R., R.K., V.E., S.J.S., M.Mitchell, M.Maurer, M.J., J.L., P.F., and M.Matsumoto are full-time employees of Arialys Therapeutics., Inc. A.K., T.K., M.Maeda, Y.A., T.Shimomura, T.M., K.K., T.Shimada, K.N., N.N., Y.K., A.S., D.Y., M.Adachi, and D.J.H. are full-time employees of Astellas Pharma Inc. K.S., A.O., H.S., and M.Matsumoto were full-time employees of Astellas Pharma Inc. at the time the part of research was conducted, but are no longer affiliated with the company. V.V. and S.B. are full-time employees of Vanadro, LLC. K.Z. and T.I. received grant support from Astellas Pharma, Inc. The remaining authors declare no competing interests. ART5803 was originally developed by Astellas Pharma, Inc. ART5803 was subsequently acquired from Astellas by Arialys Therapeutics, Inc. via the execution of an exclusive worldwide licensing agreement. Arialys further developed the asset, including work within this manuscript, and is now conducting clinical studies. The work described in this manuscript was therefore conducted under the control of either Astellas or Arialys.
