## [Transparent Peer Review file · Nature Communications]

Monoclonal humanized monovalent antibody blocking therapy for anti-NMDA receptor encephalitis

Corresponding Author: Dr Mitsuyuki Matsumoto

Version 0:

Reviewer comments:

Reviewer #4

(Remarks to the Author)

The manuscript "Monoclonal humanized one-armed antibody blocking therapy for anti-NMDA receptor encephalitis" by Kanno et al. is a well-conducted and original study. The authors developed an effectorless, one-armed human IgG monoclonal antibody with high affinity towards the NR1 domain of the NMDAR, which is also the target of most autoantibodies found in patients with NMDAR encephalitis.

In this study, the authors propose a rich set of novel data that together aim to characterize this recombinant antibody biochemically, in vitro and on organotypic slides, and explore its potential for usage as a therapeutic tool to block NMDA receptor internalization by pathogenic antibodies in a marmoset NMDAR encephalitis model.

After one round of revisions already carried out, the manuscript is sound, well-written and of significant interest to the field.

I have just a few comments:

1) Even the slightest modifications of the IgG backbone may induce an immune reaction upon injection. As the authors suggest here to use the ART5803 modified antibody for therapeutic use over longer periods (weeks), it is important to demonstrate that the mutations introduced into the human IgG backbone do not elicit an anti-ART5803 ab response in their marmoset model. This could be achieved by comparing anti-ART5803 ab responses to responses against an unmutated hIgG1.

Minor:

2) It is stated in lines 372/273 that continuous infusion of ART5803 alone did not induce any noticeable behavioral changes in marmoset. Please include these data.

3) In line with this data, Figure 1i and indirectly also 1k seem to indicate that ART5803 may rather stabilize the NMDAR on the cell surface. Did the authors evaluate this possibility?

4) The use of an unpaired student's t-test for the statistical analysis of n=3 independent replicates seems inappropriate.

Reviewer #5

(Remarks to the Author)

In the study, Kanno and colleagues generated a novel one-armed humanized IgG1 with a silent Fc in order to compete with the binding of pathogenic autoantibodies directed against the NMDA receptor (NMDAR). The authors provide evidence that ART5803 binds with high affinity to the same region of a well-characterized monoclonal antibody. The authors then tested the capacity of ART5803 to prevent cellular and behavioural effects of monoclonal and patients autoantibodies. A new model with marmoset was then used to show the in vivo relevance of ART5803. The topic of the study is of great interest and the need for such a tool for therapeutical use is also of interest and timely. However, the study suffers major flaws, both in the experimental design, description of experiments, analysis and even knowledge of the current literature. Although the topic is of great interest and the use of marmoset is interesting, the manuscript in its current form is at best a preliminary report, poorly presented. Some of the major flaws are described below.

- i) the study is based on the use of a novel tool, ART5803. However, there is simply no description of how ART5803 was selected and prepared. The basic characterization provided to the reader is way to simplistic.
- ii) Among all the missing key controls, I will only mention one. All the experiments need to be performed with an additional « two-armed » ART5803. None of the conclusions raised by the authors are conclusive without such a control. Similarly, the Fab-ART5803 should also be tested. The fact that a single arm antibody acts in such a way is not sufficient to demonstrate a monovalency/bivalency model.
- iii) All the experiments are poorly presented, which is a least to say. The reader only has a graph without any representation of the actual data. Each series of experiments is somehow a guess for the reader on how exactly the experiments were performed.
- iv) As an example of the above point, the experiments performed and presented in Fig 2b are more than troublesome. The extremely poor quality of the images raise the concern on whether the whole experiment has been properly performed. Here again, we miss all the requested controls to ensure that GluN-SEP fluorescence come from surface NMDA receptor. What was measured ? The spine fluorescence ? The whole fluorescence, shaft and spine ? based on the quality of the image, the whole dataset should be strongly questioned...
- v) The whole manuscript is full of inaccurate statements, such as « there are no approved treatments » for NMDA receptor encephalitis... The NMDA receptor subunit nomenclature is not the correct one... Recent papers have described the binding of 003-102 autoantibody, and its putative ionotropic action (Michalski et al., NSMB, 2024), and another one has shown that anti-NMDA receptor autoantibodies do not primarily act as cross-linker (Jamet et al., Brain, 2024). The authors are strongly encouraged to update their bibliography and compared their data with other similar reports. These aspects further substantiate the preliminary status of this manuscript. The manuscript would strongly benefit from a thorough editing.

Version 1:

Reviewer comments:

Reviewer #4

(Remarks to the Author)

The authors have adequately addressed the points raised by the reviewers. I nevertheless insist that a point should be added to the discussion of the paper referring to the very low sample size in some of the experiments and the hence very limited meaning of a t-test for statistical analysis.

Reviewer #5

(Remarks to the Author)

The authors did not, or very partially, address my concerns.

First, bivalent ART5803 experiments are indeed of great importance and the data in Fig2c are of support. However, the main claim of the paper, which include different scales, should be properly addressed with this bivalent ART5803. At least one key functional assay.

Second, the authors mention they have addressed the quality of the experiments and images. This is simply not respectful to the reviewer. If Fig3b images are the best examples to sustain a key claim of the paper, then we have different definition of what is a high-standard experiment. I do not see any changes in the poor quality images, nothing! Yet, I should simply accept the whole conclusion. I won't further comment.

Version 2:

Reviewer comments:

Reviewer #5

(Remarks to the Author)

The authors have addressed part of my critics, adding some experiments but missing other key ones. The manuscript is clearly improved.

Response to Reviewers' Comments

We truly appreciate reviewers' valuable feedback and have revised the manuscript accordingly. The edited parts in the manuscript have been highlighted. Below are our detailed responses to the comments and suggestions.

Reviewer #4

1) Even the slightest modifications of the IgG backbone may induce an immune reaction upon injection. As the authors suggest here to use the ART5803 modified antibody for therapeutic use over longer periods (weeks), it is important to demonstrate that the mutations introduced into the human IgG backbone do not elicit an anti-ART5803 ab response in their marmoset model. This could be achieved by comparing anti-ART5803 ab responses to responses against an unmutated hlgG1.

→ We appreciate this thoughtful observation. Because the human IgG1 backbone sequence is slightly different from marmoset IgG1, we agree that ART5803 (human IgG1 backbone) could be immunogenic in marmosets. In the marmoset IP efficacy study, however, we did not see any obvious pharmacokinetic disturbances by potential ADA. In humans, LALA mutations and "knob in hole" mutations have been reported to not be immunogenic. We have added two references (Tsai et al., 2024, Abdeldaim and Schindowski, 2023) where these mutations and immunogenicity were discussed. We understand that ADA assessment is very important in humans. We are currently testing ADA against ART5803 in humans in a Phase 1 clinical study.

Minor:

2) It is stated in lines 372/273 that continuous infusion of ART5803 alone did not induce any noticeable behavioral changes in marmoset. Please include these data.

→ We have added Supplementary Table 6 (daily evaluation of clinical signs indicating abnormal behavior) in marmosets with IP injections of ART5803 alone 200, 400 and 800 mg/kg for 2 weeks. 400 mg/kg IP reached 1 – 2 µg/mL in CSF. As K_D of ART5803 is 0.7 nM = 0.07 µg/mL, they were over-saturated concentrations for NMDAR binding and enough to say no abnormal behaviors caused by ART5803 binding to NMDARs in the brain. Because the ICV infusion

test of ART5803 alone was performed in our preliminary study, we do not have a table to show abnormal behaviors day by day (we only reported that we did not see any noticeable behavior changes within two weeks).

3) In line with this data, Figure 1i and indirectly also 1k seem to indicate that ART5803 may rather stabilize the NMDAR on the cell surface. Did the authors evaluate this possibility?

→ We appreciate this keen observation. In NMDAR expressing HEK293 cells, NMDARs (GluN1/GluN2B) were overexpressed to test NMDAR internalization. Due to that overexpression, we saw a small increase over 100% (+ 5 - 10%) from ART5803 treated cells at higher concentrations over its Kd (0.7 nM = 0.07 µg/mL), presumably due to by suppression of a fraction of overexpressed NMDAR internalization/turnover on HEK293 cells. We have now retested NMDAR internalization with ART5803 and isotype control separately. In this experiment, we saw a general small increase of GluN1 levels by ART5803, however, this effect was also observed with the isotype control (New Figure 2b). In our ex vivo mouse hippocampal organotypic culture study, we did not see such an obvious increase of NMDARs over 100% by ART5803 incubation alone (New Figure 3e). We have considered the intriguing possibility of ART5803 as an NMDAR stabilizer as you suggested, but from current datasets, it may be premature to propose it.

4) The use of an unpaired student's t-test for the statistical analysis of n=3 independent replicates seems inappropriate.

→ We understand your concern. The ICV study was our first attempt, and due to lack of marmoset availability, the sample size was limited. Although it was not ideal, we performed student's t-test with N=3 (technically acceptable). Because of the ICV study limitations, we designed the IP efficacy study using a more appropriate N in the vehicle group.

Reviewer #5 (Remarks to the Author):

i) the study is based on the use of a novel tool, ART5803. However, there is simply no description of how ART5803 was selected and prepared. The basic characterization provided to the reader is way to simplistic.

→ We appreciate this input and have corrected this oversight on our part. We have now added a more detailed description of how ART5803 was selected and prepared in the Results section.

ii) Among all the missing key controls, I will only mention one. All the experiments need to be performed with an additional « two-armed » ART5803. None of the conclusions raised by the authors are conclusive without such a control. Similarly, the Fab-ART5803 should also be tested. The fact that a single arm antibody acts in such a way is not sufficient to demonstrate a monovalency/bivalency model.

→ We have now performed an additional study and tested bivalent (two-armed) ART5803 and Fab ART5803 in an NMDAR internalization assay. Importantly, we now show in New Figure 2c that bivalent (two-armed) ART5803 strongly induced NMDAR internalization. We believe it is sufficient to conclude bivalent (two-armed) ART5803 and ART5803 (monovalent, one-armed) performed oppositely in this experiment. Furthermore, ART5803 behaves similar to Fab ART5803. These data strongly support the validity of the monovalency/bivalency model.

→ As this manuscript focuses on assessment of the therapeutic potential of ART5803 for anti-NMDAR encephalitis patients, and repeating all the other experiments would take many years, it is not feasible to retest using bivalent and Fab ART5803 in all of the assays in the manuscript. We believe this NMDAR internalization study is sufficient to show the mechanistic difference between ART5803 (or its Fab) and “bivalent ART5803” on the receptor.

iii) All the experiments are poorly presented, which is a least to say. The reader only has a graph without any representation of the actual data. Each series of experiments is somehow a guess for the reader on how exactly the experiments were performed.

→ We regret this oversight. Because most experiments were carried out on the large scale in multi-well plates, the primary data is in the form of fluorescence amplitudes only, without any spatial information. We added New Figure 2a to show the data of NMDAR internalization (close to the actual data) and New Supplementary tables 5 and 7 to show the data for marmoset studies.

iv) As an example of the above point, the experiments performed and presented in Fig 2b are more than troublesome. The extremely poor quality of the images raise the concern on whether the whole experiment has been properly performed. Here

again, we miss all the requested controls to ensure that GluN-SEP fluorescence come from surface NMDA receptor. What was measured ? The spine fluorescence ? The whole fluorescence, shaft and spine ? based on the quality of the image, the whole dataset should be strongly questioned...

- We regret the poor quality of the images from these experiments in the previous version. We realize now that they had inadvertently been passed from PDF through PowerPoint, pasted into Word and then converted back to PDF, which destroys the image quality. We are now submitting the original PDF version without such processing and are confident that the images will meet journal publication standards. At the same time, it is still important to note that SEP-GluN2A is expressed at low levels, and thus the images will be much noisier than for a highly expressed GFP-tagged protein like PSD-95, or for a GFP cell fill.
- Regarding the quantification of SEP-GluN2A, it was previously detailed that spine fluorescence was measured and how in the Methods section under “Image Analysis”. We have now added to the Results section that spine fluorescence was quantified. SEP-GluN2A has been well-established as a surface marker, as in the original paper from which the constructs originated (Kopeck et al., 2007), which is now referred to when we introduce the construct.

v) The whole manuscript is full of inaccurate statements, such as « there are no approved treatments » for NMDA receptor encephalitis... The NMDA receptor subunit nomenclature is not the correct one... Recent papers have described the binding of 003-102 autoantibody, and its putative ionotropic action (Michalski et al., NSMB, 2024), and another one has shown that anti-NMDA receptor autoantibodies do not primarily act as cross-linker (Jamet et al., Brain, 2024). The authors are strongly encouraged to update their bibliography and compared their data with other similar reports. These aspects further substantiate the preliminary status of this manuscript. The manuscript would strongly benefit from a thorough editing.

- We appreciate this input and have now thoroughly reworked the manuscript, including the following adjustments:
 - We have reworded the phrase “there are no approved treatments” to “there is no regulatory approved treatment for anti-NMDAR encephalitis”.
 - We have changed NR1/NR2 to GluN1/GluN2 to incorporate appropriate NMDA receptor subunit nomenclature.
 - We have added several additional key references, including Jamet et al., Brain, 2024 and Michalski et al., NSMB, 2024.

- We have discussed the recently observed direct effects of pathogenic autoantibodies on NMDAR function other than crosslinking-internalization and how that relates our observations with ART5803 in the discussion.

Response to Reviewers' Comments

We appreciate reviewers' valuable feedback and have revised the manuscript accordingly. The edited parts in the manuscript have been highlighted. Below are our detailed responses to the comments and suggestions.

Reviewer #4

The authors have adequately addressed the points raised by the reviewers. I nevertheless insist that a point should be added to the discussion of the paper referring to the very low sample size in some of the experiments and the hence very limited meaning of a t-test for statistical analysis.

- We express our appreciation to reviewer #4 for the additional input and guidance to help us improve the paper.
- We have now added a sentence to describe the limitation of statistical analysis of some studies with small N in the Discussion as below.

(Very end of the Discussion)

Additionally, ART5803 may be a unique treatment option for relapse and/or treatment resistant conditions caused by sustained anti-NMDAR autoantibody production. The clinical potential of ART5803 will be addressed in future clinical trials.

Although scarcity of marmosets and samples from a rare patient population restricted the sample size and robust statistical analyses in some studies, given the profound and rapid efficacy of ART5803 observed in our marmoset model, we believe that ART5803 could become a promising therapy for patients with anti-NMDAR encephalitis and potentially for other CNS disorders caused by anti-NMDAR autoantibodies.

Reviewer #5 (Remarks to the Author):

The authors did not, or very partially, address my concerns.

- We made significant efforts to address the reviewer's concerns and revised the manuscript accordingly. We regret that the reviewer feels his/her concerns were not fully addressed. We believe that the revision based on the reviewer's comments/concerns made our manuscript more compelling and convincing.

First, bivalent ART5803 experiments are indeed of great importance and the data in Fig2c are of support. However, the main claim of the paper, which include different scales, should be properly addressed with this bivalent ART5803. At least one key functional assay.

- In our NMDAR internalization assay, we confirmed bivalent ART5803 induced NMDAR internalization in a similar way to pathogenic #003-102 Ab. This result is in contrast to the lack of NMDAR internalization observed in response to ART5803. We believe the NMDAR internalization study was one key functional assay. Furthermore, our HDX-MS study demonstrated that ART5803 and #003-102 Ab bind to an almost identical epitope, which supports the similar behavior shared with #003-102 Ab and bivalent ART5803. From these data, we believe that #003-102 Ab and bivalent ART5803 should behave similarly as a pathogenic antibody.
- Importantly, our manuscript highlights and investigates the suitability of ART5803 as a potential therapeutics rather than the lack of suitability of bivalent ART5803. Considering our funding status and timing of our ART5803 program, it is not feasible to retest using bivalent ART5803 in other studies.

Second, the authors mention they have addressed the quality of the experiments and images. This is simply not respectful to the reviewer. If Fig3b images are the best examples to sustain a key claim of the paper, then we have different definition of what is a high-standard experiment. I do not see any changes in the poor quality images, nothing! Yet, I should simply accept the whole conclusion. I won't further comment.

- We regret that we erroneously attributed the reviewer's original criticisms to image processing errors during the original manuscript submission. We have considerable experience with quantitative fluorescence image analysis and, while SEP-tagged GluN indeed expresses at such low levels that changes of 30% are challenging to discern by visual inspection alone, software-based analysis, as described in the methods, confirms these changes are significant. Upon the initial round of review we did revisit our analysis with an independent researcher who confirmed that the results are robust as originally reported. We have now thoroughly examined our data to select new ROIs for 4 of 5 conditions. We hope that these images now provide a better representation of the results and adequately address the reviewer's concerns.

Response to Reviewers' Comments

We appreciate reviewers' valuable feedback.

Reviewer #5 (Remarks to the Author):

The authors have addressed part of my critics, adding some experiments but missing other key ones. The manuscript is clearly improved.

→ We are grateful for the reviewer's thorough and rigorous feedback throughout this process. We have made substantial improvements to the manuscript in response to the reviewer's comments and have implemented additional experiments where possible. Although not every suggestion could be fully addressed due to practical constraints, we believe the manuscript is now in a much stronger position.